# Multi-View Encoders for Performance Prediction in LLM-Based Agentic Workflows

**Patara Trirat[1], Wonyong Jeong[1], Sung Ju Hwang[1 2]**
[1]DeepAuto.ai, [2]KAIST
{patara, young, sjhwang}@deepauto.ai

## Abstract

Large language models (LLMs) have demonstrated remarkable capabilities across diverse tasks, but optimizing LLM-based agentic systems remains challenging due to the vast search space of agent configurations, prompting strategies, and communication patterns. Existing approaches often rely on heuristic-based tuning or exhaustive evaluation, which can be computationally expensive and suboptimal. This paper proposes **Agentic Predictor**, a lightweight predictor for efficient agentic workflow evaluation. Agentic Predictor is equipped with a *multi-view workflow encoding* technique that leverages multi-view representation learning of agentic systems by incorporating code architecture, textual prompts, and interaction graph features. To achieve high predictive accuracy while significantly reducing the number of required workflow evaluations for training a predictor, Agentic Predictor employs *cross-domain unsupervised pretraining*. By learning to approximate task success rates, Agentic Predictor enables fast and accurate selection of optimal agentic workflow configurations for a given task, significantly reducing the need for expensive trial-and-error evaluations. Experiments on a carefully curated benchmark spanning three domains show that our predictor outperforms several strong graph-based baselines in both predictive accuracy and workflow utility, highlighting the potential of performance predictors in streamlining the design of LLM-based agentic workflows.

## 1 Introduction

Large language models (LLMs) have catalyzed the development of agentic systems capable of executing complex, multi-step tasks autonomously (Hong et al., 2024; Wu et al., 2024; Xi et al., 2024; Mialon et al., 2024). These systems, often constructed through meticulous manual engineering, integrate components such as Chain-of-Thought reasoning, tool invocation, and memory management to enable sophisticated behaviors for orchestrating intricate workflows (Xi et al., 2025; Ke et al., 2025; Gridach et al., 2025; Plaat et al., 2025). However, the handcrafted nature of these systems imposes limitations on scalability, adaptability, and rapid deployment across diverse domains.

To address these limitations, recent trends have shifted towards automated design methods for agentic systems (Hu et al., 2025a; Shang et al., 2025; Zhang et al., 2025b; Zhuge et al., 2024; Liu et al., 2024b; Hu et al., 2025b; Yuan et al., 2025). Automated methods typically employ search algorithms to discover optimal workflow configurations by systematically exploring a vast design space. Instead of relying on human intuition, these approaches generally involve iterations of candidate generation, evaluation, and refinement. While promising, these methods exhibit significant drawbacks, chiefly the high computational costs associated with the extensive validation steps needed during the exploration and evaluation phases of the search. Each candidate configuration must undergo rigorous evaluation, often through expensive, repeated interactions with LLM APIs, rendering the search prohibitively costly and time-consuming.

In this paper, we argue that purely search-based automated design methods are inherently inefficient and propose a predictive approach to significantly accelerate workflow evaluation. Specifically, we advocate for a predictor-based framework that can rapidly estimate the performance of candidate agentic workflows, similar to performance predictors in neural architecture search (White et al., 2021), thereby reducing the need for extensive validation.

As depicted in Figure 1, instead of fully evaluating every candidate, a predictive model can estimate the quality and viability of agentic workflows, thus guiding the search process far more efficiently. By reducing costly ground-truth executions or environment interactions during the search process, prediction-based approaches promise significant improvements in both search efficiency and solution quality. However, building a high-quality predictor for agentic workflows introduces two fundamental challenges.

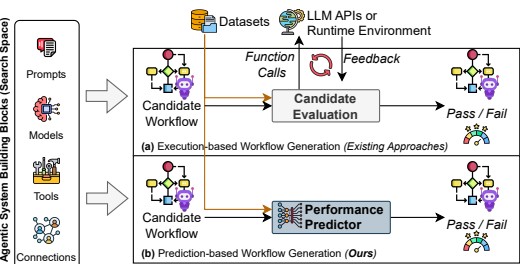

Figure 1: Comparison between (a) execution-based and (b) prediction-based candidate evaluation for agentic workflow generation. Execution-based methods rely on costly runtime or LLM calls, while our prediction-based approach offers faster, scalable evaluation via a learned predictor.

**Workflow Heterogeneity**. Agentic workflows exhibit considerable heterogeneity; subtle variations in configuration can lead to dramatically different performances. Specifically, workflows can vary widely in communication structure, prompting strategies, tool invocation patterns, and reasoning styles, making it challenging to learn a unified predictive model. Moreover, agentic systems differ significantly across tasks, domains, and toolsets, resulting in diverse and complex workflow configurations that are difficult to model uniformly (Xu et al., 2024; Qiao et al., 2025).

**Scarcity of Labeled Data**. The availability of labeled data for training effective prediction models is severely limited due to the prohibitive cost of generating performance labels through exhaustive validation. Constructing a large, diverse set of labeled workflows with known execution outcomes is particularly expensive, creating a data bottleneck for supervised learning approaches. Moreover, gathering large-scale, high-quality labels for agentic workflows (e.g., success rates and execution outcomes) is often infeasible, further limiting the amount of supervised training data available for learning accurate predictors.

To tackle these challenges, we present **Agentic Predictor**, a multi-view encoder framework for performance prediction in LLM-based agentic workflows. To address workflow heterogeneity, Agentic Predictor uses *multi-view workflow encoders* that jointly model complementary signals—structural (e.g., agent topology), behavioral (e.g., tool usage), and semantic (e.g., prompts)—capturing the diverse, task-dependent characteristics of workflow configurations. To mitigate label scarcity, we introduce *cross-domain unsupervised pretraining*, denoted Agentic Predictor+, which leverages abundant unlabeled workflows from related domains. We pretrain the multi-view encoders with contrastive and reconstruction objectives, then fine-tune on limited labeled data, yielding robust and transferable representations for prediction. The **main contributions** of this paper are as follows.

- We propose multi-view encoders and cross-domain unsupervised pretraining that jointly capture the heterogeneous facets of LLM-based agentic workflows, yielding higher predictive performance, better generalization, and effective predictor training under limited labels.
- We introduce Agentic Predictor, unifying these components for the underexplored problem of performance prediction in heterogeneous, label-scarce LLM-based agentic workflows, thereby reducing trial-and-error costs and accelerating development.
- We empirically demonstrate that, averaged across three domains, Agentic Predictor improves prediction accuracy by up to **6.90%** and utility by up to **5.87%** over strong baselines.

## 2 RELATED WORK

**Automated Generation of Agentic Workflows**. Recent advancements (Xi et al., 2025; Ke et al., 2025; Gridach et al., 2025; Plaat et al., 2025) in agentic workflows have led to the development of various frameworks aimed at enhancing multi-agent collaboration for complex tasks (Guo et al., 2024; Trirat et al., 2025; Niu et al., 2025; Trirat & Lee, 2025a). MetaGPT (Hong et al., 2024) and ChatDev (Qian et al., 2024) use predefined multi-agent structures to address coding challenges, while AgentVerse (Chen et al., 2024) introduces iterative collaboration where agents discuss, execute, and evaluate tasks. LLM-Debate (Du et al., 2024) employs multiple expert agents that engage in debates over several rounds to derive final answers. However, these systems often rely on static configurations, which limits their adaptability to diverse queries across different tasks and domains.

To optimize agentic workflows, GPTSwarm (Zhuge et al., 2024) and G-Designer (Zhang et al., 2025a) apply variants of the REINFORCE algorithm to optimize workflow structures represented as directed acyclic graphs (DAGs), while DyLAN (Liu et al., 2024b) dynamically selects agents based on task requirements. ADAS (Hu et al., 2025a) and AFlow (Zhang et al., 2025b) further leverage powerful LLMs (e.g., Claude-3.5-Sonnet and GPT-4) to iteratively generate task-specific multi-agent systems. Similarly, AgentSquare (Shang et al., 2025) proposes a modular design space for automatic LLM agent search, enhancing adaptability to novel tasks. Despite their effectiveness, these methods typically require numerous LLM calls, resulting in significant computational and financial overheads, making them less practical for real-world applications.

Rather than manually designing a fixed workflow (Qian et al., 2024; Chen et al., 2024; Du et al., 2024) or paying repeated inference costs to synthesize one per query (Zhuge et al., 2024; Liu et al., 2024b), Agentic Predictor presents a lightweight

Table 1: Comparison between ours and existing frameworks for prediction-based workflow generation.

| Framework | Multi-View Representation | Unsupervised Pretraining | Lightweight Predictor | Search Agnostic |
|---|---|---|---|---|
| MAS-GPT (Ye et al., 2025) | ✗ | ✗ | ✗ | ✗ |
| FLORA-Bench (Zhang et al., 2025c) | ✗ | ✗ | ✓ | ✓ |
| **Agentic Predictor** *(Ours)* | ✓ | ✓ | ✓ | ✓ |

performance predictor to rapidly estimate the quality of candidate agentic workflows, enabling broad exploration without exhaustive evaluations. Among recent efforts, FLORA-Bench (Zhang et al., 2025c) advocates GNN-based predictors and releases a benchmark that models workflows as a single-view graph where prompts are node features. A complementary direction, MAS-GPT (Ye et al., 2025), fine-tunes LLMs to directly generate workflows in a single call. **In contrast to these directions**, Agentic Predictor differs in three respects: *representation* (multi-view encoding of agent topology, code, and system prompts vs. single-view graphs), *learning* (cross-domain unsupervised pretraining to mitigate label scarcity, rather than no pretraining recipe or supervised LLM fine-tuning), and *efficiency* (a compact predictor for fast evaluation without repeated LLM calls). Table 1 summarizes these distinctions.

**Performance Predictors for NAS**. Neural architecture search (NAS) has spurred the development of performance predictors that aim to reduce the significant computational cost of evaluating candidate architectures. PRE-NAS (Peng et al., 2022) employs a predictor-assisted evolutionary strategy to estimate model performance, thereby alleviating the need for exhaustive training. BRP-NAS (Dudziak et al., 2020) integrates graph convolutional networks to forecast hardware-aware performance metrics, improving the practicality of NAS under resource constraints. CAP (Ji et al., 2024) introduces a context-aware neural predictor, leveraging self-supervised learning to generate expressive and generalizable representations of architectures, thus enabling more effective search space exploration. FlowerFormer (Hwang et al., 2024) advances architecture encoding through a flow-aware graph transformer, yielding improved prediction accuracy. A unifying trend among these methods is the emphasis on learning more *informative representations* to guide the search process. Building on this insight, we propose the Agentic Predictor framework, which approaches performance prediction from a *representation-centric* perspective. By incorporating multi-view representations conditioned on workflow configurations, Agentic Predictor facilitates accurate performance estimation and efficient exploration of the agentic workflow space.

## 3 METHODOLOGY: AGENTIC PREDICTOR

### 3.1 PROBLEM FORMULATION

Let an agentic workflow be denoted as $\mathcal{W} = \{\mathcal{V}, \mathcal{E}, \mathcal{P}, \mathcal{C}\}$, where $\mathcal{V} = \{v_i\}_{i=1}^N$ represents the set of $N$ agents, $\mathcal{E}$ denotes the set of edges defining the connections between agents, and $\mathcal{P} = \{p_i\}_{i=1}^N$ denotes the system prompts for each agent $i$. $\mathcal{C}$ represents the complete code specifying the logic and structure of the workflow. Thus, the workflow $\mathcal{W}$ is represented as a directed acyclic graph (DAG).

Given a task description $T$, the workflow $\mathcal{W}$ autonomously executes agents in topological order, where the $i$-th agent receives the task description $T$ along with the outputs $y$ from its predecessor agents. Formally, the input to agent $i$ is defined as $\mathcal{X}_i = \{T\} \cup \{y_j : v_j \in \mathcal{N}_i^{(\text{in})}\}$, where $\mathcal{N}_i^{(\text{in})}$ denotes the set of predecessor agents of agent $i$, and $y_j$ is the output of agent $j$. The output $y_i$ of agent $i$ is generated by querying an LLM: $y_i = \text{LLM}(\mathcal{X}_i, p_i)$. After executing all agents, the final response of the agentic workflow is defined as $r = f_{\text{LLM}}(\mathcal{W}, T)$, where $f_{\text{LLM}}$ represents the overall execution process of the given LLM. Generally, this process is repeated for the evaluation of $r$, which incurs significant computational and financial overhead.

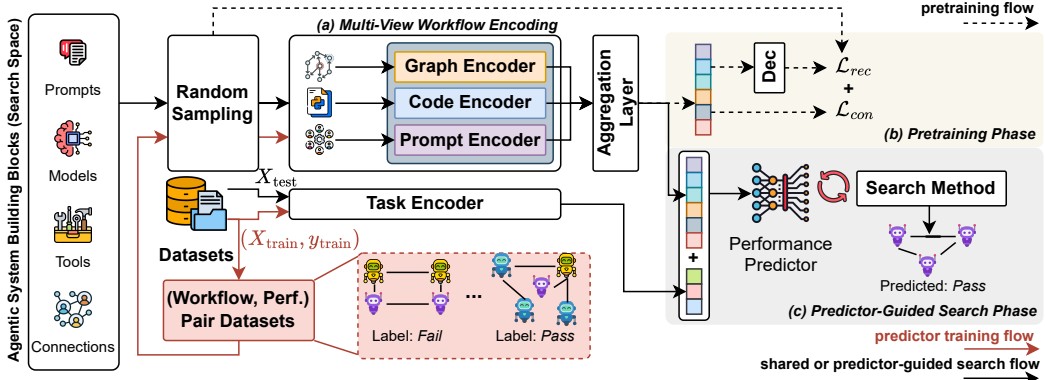

Figure 2: Overview of our Agentic Predictor framework. A **(a) multi-view workflow encoder** is designed to encode a set of agentic workflows from graph, code, and prompt aspects into unified representations, which serve as features for training the predictor. In the **(b) pretraining phase**, the encoder learns these representations on unlabeled workflows spanning diverse tasks and domains, using cross-domain unsupervised pretraining objectives. In the **(c) predictor-guided search phase**, a performance predictor is trained on a small (workflow configuration, performance) dataset to classify configurations as pass or fail, and subsequently guides the search toward promising configurations.

In contrast, this paper aims to design a predictive model $\mathcal{M}$ that efficiently estimates the final performance of an agentic workflow $\mathcal{W}$ on a given task description $T$, without requiring costly LLM invocations. Therefore, we treat the workflow $\mathcal{W}$ and task description $T$ as inputs to the predictor $\mathcal{M}$, which outputs the estimated performance $\hat{e}$. Formally, $\hat{e} = \mathcal{M}_{\Theta}(\mathcal{W}, T)$, where $\Theta$ denotes the learnable parameters of the performance predictor.

**Learning Objective**. Given the workflow $\mathcal{W}$, task description (or query) $T$, and performance predictor $\mathcal{M}_{\Theta}$ parameterized by $\Theta$, we aim to find the optimal $\Theta$ that minimizes the error between the estimated performance $\hat{e}$ and the ground-truth performance $e$. Formally, we solve

$$\min_{\Theta} \mathbb{E}_{(\mathcal{W}, T)}[\mathcal{L}(e, \hat{e})], \tag{1}$$

where $\mathcal{L}(\cdot, \cdot)$ is a loss function that quantifies the discrepancy between the ground truth and the predicted performance. $\mathcal{L}$ can be either the cross-entropy loss or the mean squared error loss, depending on whether the prediction task is formulated as a classification or regression problem.

### 3.2 FRAMEWORK OVERVIEW

We present an overview of our Agentic Predictor framework in Figure 2. First, the multi-view workflow encoder integrates key aspects of an agentic workflow—graph structures $(\mathcal{V}, \mathcal{E})$, code implementations $\mathcal{C}$, and system prompts $\mathcal{P}$—into a unified representation $\mathcal{F}$. Integration is achieved via modality-specific encoders followed by an aggregation layer that consolidates features across modalities. Second, when labeled instances are scarce, we refine these representations with unsupervised objectives, reconstruction and contrastive learning, to improve generalization and adaptability across diverse tasks and configurations. Third, a dedicated performance predictor $\mathcal{M}_{\Theta}$ is trained on a labeled set (often small) comprising workflow configurations $\mathcal{W}$, task descriptions $T$, and observed performance outcomes $e$. Finally, with the trained predictor, we perform a predictor-guided search that efficiently ranks and selects promising workflow configurations without incurring expensive LLM calls. Because Agentic Predictor is search-agnostic, we deliberately do not commit to a specific search algorithm within the framework.

### 3.3 MULTI-VIEW WORKFLOW ENCODING

Motivated by recent findings in the NAS literature (White et al., 2020; Akhauri & Abdelfattah, 2024; Trirat & Lee, 2025b), which show that architecture representations strongly influence predictor performance, we argue for expressive, comprehensive representations tailored to agentic workflows. Because agentic workflows differ fundamentally from traditional neural architectures, conventional

graph-based encodings alone are insufficient. Although DAGs naturally capture explicit inter-agent communication and dependencies, they omit crucial implicit signals such as tool-usage patterns, code structure, computational complexity, and the nuanced semantics present in agent prompts. To address these limitations, we propose a multi-view encoding scheme that integrates complementary representations at multiple granularities, with each view capturing distinct yet essential aspects of LLM-based agentic workflows.

- **Graph View** explicitly models structural dependencies and direct interactions among agents, emphasizing inter-agent communication channels. We denote the graph view as $\mathcal{G} = (\mathcal{V}, \mathcal{E})$.
- **Code View** implicitly encodes program-level semantics, control and logical sequence, computational complexity, and patterns of tool usage inherent in workflow implementations $\mathcal{C}$.
- **Prompt View** provides semantic embeddings that capture agent roles, behavioral specifications, and broader contextual guidance embedded within system and instruction prompts $\mathcal{P}$.

Our rationale for adopting this multi-view framework is that aggregating heterogeneous information sources reduces representation bias, thereby improving both robustness and predictive accuracy.

### 3.3.1 ENCODER NETWORKS

We now detail the components of our proposed multi-view workflow encoding method for performance prediction in agentic workflows using neural networks. Let $\text{Enc}(\cdot)$ denote an encoder function that maps a candidate workflow—composed of $(\mathcal{G}, \mathcal{C}, \mathcal{P})$—into $d$-dimensional Euclidean space, i.e., $\text{Enc}(\cdot) : (\mathcal{G}, \mathcal{C}, \mathcal{P}) \rightarrow \mathbb{R}^d$. Given the heterogeneous nature of workflow configurations, we design three specialized encoder networks, each responsible for learning a representation corresponding to a distinct view. These view-specific representations are then aggregated into a shared latent space, denoted by $\mathbf{Z} = \text{Enc}(\mathcal{G}, \mathcal{C}, \mathcal{P})$, where $\mathbf{Z} \in \mathbb{R}^d$. This continuous latent representation is used to train the performance predictor $\mathcal{M}_\Theta$ (see §3.5). The individual encoders for each view are integrated into a unified architecture as described below.

**Graph Encoder**. We employ graph neural network (GNN) layers to encode graph-based representations. The workflow is modeled as a DAG in which each edge encodes a unidirectional message channel. Rather than relying on a single graph, we adopt a *multi-graph* approach that integrates node features from multiple views, including agent-specific definitions and function-call implementations at each agent node. We instantiate this multi-graph representation with three graph views constructed from a workflow $\mathcal{W}$. In the *prompt graph* $\mathcal{G}_{\text{prompt}}$, node features are obtained by pooling the embeddings of each agent's system and instruction prompts. In the *code graph* $\mathcal{G}_{\text{code}}$, node features correspond to the function-call code associated with each agent. In the *operator graph* $\mathcal{G}_{\text{operator}}$, node features encode operator types and their definitions. All three graphs share the same node and edge set, where edges can be derived from an abstract syntax tree as suggested by Zhang et al. (2025c).

We then obtain view-specific node embeddings $\mathbf{H}_{\text{prompt}} = \text{GNN}(\mathcal{G}_{\text{prompt}})$, $\mathbf{H}_{\text{code}} = \text{GNN}(\mathcal{G}_{\text{code}})$, and $\mathbf{H}_{\text{operator}} = \text{GNN}(\mathcal{G}_{\text{operator}})$ in $\mathbb{R}^{N \times d}$, stack them along a view dimension to form $\mathbf{X} \in \mathbb{R}^{N \times V \times d}$ with $V = 3$, and apply a cross-view self-attention block with residual connection and layer normalization $\hat{\mathbf{X}} = \text{LN}(\text{MHA}(\mathbf{X}, \mathbf{X}, \mathbf{X}) + \mathbf{X})$, where MHA is multi-head attention applied across views for each node (the sequence axis is the view axis; topology is unchanged).

Next, a view-attention pooling module computes per-node attention weights with a $L$-layer multi-layer perceptron (MLP) and $\tanh$ nonlinearity, followed by a softmax over views, and produces a weighted sum across views $\mathbf{H} = \text{ViewAttnPool}(\hat{\mathbf{X}}) \in \mathbb{R}^{N \times d}$. Finally, a graph readout $G_{\text{pool}}$ aggregates node embeddings into a single graph representation, $\mathbf{Z}_{\mathcal{G}} = G_{\text{pool}}(\mathbf{H}) = G_{\text{pool}}(\text{ViewAttnPool}(\text{CrossGraphAttn}(\mathbf{X})))$, which preserves edge directionality from the upstream GNN while capturing cross-view contextual dependencies at the node level before graph-level summarization. Here, $\text{CrossGraphAttn}$ enriches each node with multi-view contextual dependencies, while $\text{ViewAttnPool}$ highlights which views are most informative.

**Code Encoder**. While $\mathcal{G}_{\text{code}}$ and $\mathcal{G}_{\text{operator}}$ primarily encode structural information derived from the different workflow graphs, this code encoder is designed to provide a complementary, holistic representation of the entire workflow code. To model the workflow-level embeddings, we use an $L$-layer MLP to extract latent semantic features, enabling the model to learn intricate computational logic and tool interactions at a *global* level. The code representation is computed as $\mathbf{Z}_{\mathcal{C}} = \text{MLP}_{\mathcal{C}}(\mathcal{C})$.

**Prompt Encoder**. Instead of encoding agent prompts solely as node-level features (Zhang et al., 2025c), we use a separate $L$-layer MLP to encode the *entire* workflow instruction prompt holistically. This approach captures role descriptions, behavioral intents, and global context—resulting in richer and more semantically informed representations. The prompt encoding is $\mathbf{Z}_\mathcal{P} = \mathrm{MLP}_\mathcal{P}(\mathcal{P})$.

**Aggregation Layer**. The representations from the graph, code, and prompt encoders—$\mathbf{Z}_\mathcal{G}$, $\mathbf{Z}_\mathcal{C}$, and $\mathbf{Z}_\mathcal{P}$—are concatenated and passed through a final MLP layer. This aggregation mechanism adaptively integrates information across all views, enabling the model to emphasize the most contextually relevant aspects. The final output of the encoder $\mathrm{Enc}(\cdot)$ is computed as $\mathbf{Z} = \mathrm{MLP}([\mathbf{Z}_\mathcal{G}, \mathbf{Z}_\mathcal{C}, \mathbf{Z}_\mathcal{P}])$.

Consequently, the aggregation layer acts as a *learnable* fusion module. The attention-based graph encoder produces $\mathbf{Z}_\mathcal{G}$ with cross-view interactions already embedded, and the downstream fusion assigns task-dependent importance to the graph-, code-, and prompt-level representations, rather than treating all views as equally informative. These encoders learn not only from different workflow perspectives but also at varying levels of granularity, specifically, at the graph level for agent interactions, the code level for logical structures, and the prompt level for agent-specific instructions.

### 3.3.2 DECODER NETWORKS

The decoder is a generative module that reconstructs $\hat{\mathcal{G}}$, $\hat{\mathcal{C}}$, and $\hat{\mathcal{P}}$ from the latent variables $\mathbf{Z}$ to encourage learning generalizable representations of agentic workflows. It consists of a stack of MLP layers. For simplicity, the decoder outputs the modality-specific *input embedding* vectors of $\mathcal{G}$, $\mathcal{C}$, and $\mathcal{P}$. Accordingly, we parameterize $\mathrm{Dec}(\cdot)$ with an MLP and define $\hat{\mathcal{G}} = \mathrm{MLP}(\mathbf{Z}_\mathcal{G})$, $\hat{\mathcal{C}} = \mathrm{MLP}(\mathbf{Z}_\mathcal{C})$, and $\hat{\mathcal{P}} = \mathrm{MLP}(\mathbf{Z}_\mathcal{P})$. This decoder is used only during pretraining for self-supervised reconstruction of modality-specific embeddings from $\mathbf{Z}$ and is not part of the encoding path at inference time.

### 3.4 CROSS-DOMAIN UNSUPERVISED PRETRAINING

In real-world scenarios, labeled performance data for agentic workflows are scarce due to costly evaluation. To enable data-efficient training without label leakage, we *optionally* adopt a two-phase strategy. Rather than directly supervising the encoder with performance labels, we first perform cross-domain unsupervised pretraining to obtain rich and generalizable workflow representations $\mathbf{Z}$. No performance labels (e.g., success/failure) are used in this stage. The resulting representations improve sample efficiency for downstream prediction, in line with observations in NAS (White et al., 2020; Yan et al., 2020; 2021; Akhauri & Abdelfattah, 2024; Trirat & Lee, 2025b). When sufficiently many labels are available, direct supervised learning of the predictor remains feasible.

**Multi-Task Pretraining**. We train the multi-view encoder on $M$ unlabeled workflow configurations by minimizing a combined loss comprising reconstruction and contrastive objectives: $\mathcal{L}_{rec} = \frac{1}{M} \sum_{i=1}^{M} \|\mathcal{G}_i - \hat{\mathcal{G}}_i\|^2 + \|\mathcal{C}_i - \hat{\mathcal{C}}_i\|^2 + \|\mathcal{P}_i - \hat{\mathcal{P}}_i\|^2$ and $\mathcal{L}_{con} = \frac{1}{M} \sum_{i=1}^{M} -\log \frac{\exp(\mathrm{sim}(\mathbf{Z}_i, \mathbf{Z}_j^+)/\tau)}{\sum_{k=1}^{M} \exp(\mathrm{sim}(\mathbf{Z}_i, \mathbf{Z}_k)/\tau)}$. Here, $\mathcal{G}_i, \mathcal{C}_i, \mathcal{P}_i$ denote the input graph, code, and prompt embeddings, respectively, while $\hat{\cdot}$ denotes reconstructions via modality-specific decoders. Notably, the graph branch reconstructs its own embedding target with a stop-gradient, whereas code and prompt/text are reconstructed in input space. The contrastive loss is instantiated *cross-modally* with in-batch sampling. For each configuration $i$, positives $(\mathbf{Z}_i, \mathbf{Z}_j^+)$ are the index-aligned embeddings of the configuration across two different views (e.g., $\mathcal{G}_i$ with $\mathcal{C}_i$), while negatives are all other configurations within the batch. We symmetrize the objective by swapping anchor/target and average it over the three view pairs $(\mathcal{G}, \mathcal{C})$, $(\mathcal{G}, \mathcal{P})$, and $(\mathcal{C}, \mathcal{P})$. This learning objective encourages the encoder to capture structure- and content-aware signals without observing performance outcomes. Thus, the total loss function is $\mathcal{L}_{enc} = \mathcal{L}_{rec} + \mathcal{L}_{con}$.

### 3.5 PERFORMANCE PREDICTOR

Following the unsupervised pretraining of the multi-view encoder, we introduce a lightweight performance predictor to guide exploration of the large agentic workflow space. This phase enables efficient identification of high-performing configurations with minimal supervision, using only a small set of labeled workflow–performance pairs. As shown in Figure 2(c), our predictor operates on learned workflow embeddings, enriched with task-specific context, to form a joint representation $\mathcal{F}$ used for performance prediction and downstream search.

**Task Encoder**. To capture task-specific characteristics, we incorporate a Task Encoder that generates high-level semantic embeddings from natural-language task descriptions. These embeddings, derived from pretrained language models (e.g., T5 or BERT), provide global context that helps differentiate tasks with similar surface form but distinct functional requirements. The task embedding is concatenated with the multi-view workflow representation, forming $\mathcal{F} = [\mathbf{Z}, \mathbf{T}]$, where $\mathbf{Z}$ is the encoded workflow and $\mathbf{T}$ is the task embedding. For workflow content itself (prompts and code), we pair this with lightweight, domain-specific encoders within the multi-view backbone to balance representational capacity with efficiency. This separation of modalities followed by fusion captures complementary compositional and contextual signals and supports generalization across heterogeneous tasks with varying operational goals and constraints.

The performance predictor is a lightweight prediction head $\mathcal{M}_\Theta$ (e.g., an MLP) trained on a limited set of labeled data $(X_{\text{train}}, y_{\text{train}})$, where each $X_{\text{train}}$ corresponds to $\mathcal{F}$ and $y_{\text{train}}$ is the performance label (e.g., binary success/failure or a scalar score). We instantiate the objective to match the label type. For binary labels, we use a binary cross-entropy loss, i.e., $\mathcal{L}_{\text{pred}} = -\frac{1}{N}\sum_{i=1}^{N}[e_i \log \hat{e}_i + (1 - e_i)\log(1 - \hat{e}_i)]$, where $\hat{e}_i$ is the predicted success probability. For numeric labels, we can use mean squared error, i.e., $\mathcal{L}_{\text{pred}} = \frac{1}{N}\sum_{i=1}^{N}(s_i - \hat{s}_i)^2$. By operating on semantically rich pretrained embeddings, the predictor attains strong accuracy in the low-data regime, enabling label-efficient search.

**Integration with Workflow Optimization**. With the trained predictor in place, we can perform predictor-guided search to efficiently explore the workflow configuration space. Rather than evaluating each configuration via full execution, we embed candidates into their joint representations $\mathcal{F}$ and score them using the predictor. The top-scoring candidates are selected for evaluation. This substantially reduces computational cost by focusing on the most promising regions of the search space. A simple yet effective instantiation of this strategy uses random search to sample $K$ workflow candidates from the full configuration space, and then ranks them using the learned predictor. We select the top-$k$ configurations, averaged across samples in the benchmark, for evaluation. This predictor-as-ranker setup transforms random search into a label-efficient guided procedure without requiring complex heuristics. Since our main contribution is the performance predictor rather than the optimization algorithm, we focus evaluation on prediction accuracy and ranking quality (i.e., workflow utility) in the following section. Additional results on workflow optimization appear in §B.6.

## 4 EXPERIMENTS

We conduct a comprehensive evaluation of the proposed Agentic Predictor framework from multiple perspectives, guided by the following questions: **(Q1)** How does Agentic Predictor perform as a predictor of agentic workflow performance compared to relevant baselines? **(Q2)** How do different design choices of Agentic Predictor affect its predictive accuracy? **(Q3)** Is the pretraining phase helpful for maintaining prediction quality under varying numbers of labels? **(Q4)** Does Agentic Predictor maintain strong predictive performance under out-of-distribution shifts? **(Q5)** How does Agentic Predictor compare against few-shot LLM-based workflow performance predictors?

### 4.1 SETUP

**Benchmarks**. To evaluate performance predictors for agentic workflows, we use FLORA-Bench (Zhang et al., 2025c), the only publicly available benchmark (to the best of our knowledge) that enumerates diverse workflows across multiple domains and LLM backbones. It spans five datasets covering three core areas: code generation (HumanEval (Chen et al., 2021), MBPP (Austin et al.,

Table 2: Summary of benchmark statistics.

| Domains | Code Generation (GD/AF) | Math (GD/AF) | Reasoning (GD/AF) |
|---|---|---|---|
| # workflows | 739 / 38 | 300 / 42 | 189 / 30 |
| Avg. # nodes | 5.96 / 6.11 | 6.06 / 5.49 | 5.97 / 6.58 |
| # tasks | 233 / 233 | 782 / 782 | 2,400 / 2,400 |
| # samples | 30,683 / 7,362 | 12,561 / 4,059 | 453,600 / 72,000 |

2021)), mathematical problem solving (GSM8K (Cobbe et al., 2021), MATH (Hendrycks et al., 2021b)), and general reasoning (MMLU (Hendrycks et al., 2021a)). Table 2 summarizes the benchmarks. Here, GD and AF denote the *independently* developed G-Designer (Zhang et al., 2025a) and AFlow (Zhang et al., 2025b) agentic frameworks, respectively. We emphasize structural and procedural diversity over raw difficulty; even for datasets often considered solved by prompting (e.g., GSM8K), predicting success across workflow variants remains nontrivial. For each dataset, we randomly split instances into training (80%), validation (10%), and test (10%) sets.

Table 3: Performance comparison between Agentic Predictor and baselines. The best and second-best results are highlighted in **bold** and underlined, respectively. GD is G-Designer, and AF is AFlow.

| Domain | CodeGD | | CodeAF | | MathGD | | MathAF | | ReasonGD | | ReasonAF | | Average | |
|---|---|---|---|---|---|---|---|---|---|---|---|---|---|---|
| Model | Accuracy | Utility | Accuracy | Utility | Accuracy | Utility | Accuracy | Utility | Accuracy | Utility | Accuracy | Utility | Accuracy | Utility |
| MLP | 83.88 (±0.04) | 76.16 (±0.03) | 78.02 (±0.59) | 73.94 (±1.35) | 63.22 (±0.30) | 64.13 (±0.44) | 73.73 (±0.31) | 69.64 (±0.29) | 71.54 (±0.09) | 62.41 (±1.67) | 78.45 (±0.08) | 88.48 (±0.63) | 74.81 (±0.24) | 72.46 (±0.74) |
| GCN | 84.23 (±0.04) | 79.31 (±0.10) | 84.35 (±0.34) | 72.73 (±1.18) | 64.12 (±0.17) | 63.03 (±0.59) | 76.19 (±0.42) | 66.52 (±1.66) | 72.22 (±0.03) | 59.18 (±0.85) | 87.12 (±0.14) | **91.82** (±0.81) | 78.04 (±0.19) | 72.10 (±0.81) |
| GAT | 85.14 (±0.25) | 79.50 (±0.14) | 84.49 (±0.56) | 76.46 (±0.91) | 64.84 (±0.96) | 62.32 (±0.93) | 76.44 (±0.61) | 66.51 (±1.28) | 72.16 (±0.03) | 59.44 (±1.06) | 87.07 (±0.08) | 89.40 (±0.68) | 78.36 (±0.42) | 72.27 (±0.83) |
| GCN-II | 83.81 (±0.07) | 78.45 (±0.74) | 83.72 (±0.40) | 77.75 (±0.98) | 63.56 (±0.74) | 66.02 (±0.10) | 75.04 (±0.31) | 64.33 (±0.47) | 72.29 (±0.09) | 59.10 (±1.02) | 87.28 (±0.14) | 89.92 (±1.90) | 77.62 (±0.29) | 72.60 (±0.87) |
| Graph Transformer | 85.24 (±0.19) | 80.20 (±0.64) | 84.71 (±0.45) | 74.09 (±0.35) | 63.25 (±0.70) | 64.97 (±0.36) | 75.45 (±0.23) | 66.48 (±0.96) | 72.26 (±0.08) | 60.92 (±1.79) | 86.93 (±0.27) | 90.60 (±1.97) | 77.97 (±0.32) | 72.88 (±1.01) |
| Dir-GNN | 84.85 (±0.11) | 79.81 (±0.69) | 83.45 (±0.41) | 76.08 (±0.92) | 63.01 (±0.54) | 64.68 (±1.66) | 76.11 (±0.65) | 67.97 (±0.16) | 74.25 (±0.12) | 62.64 (±0.92) | 86.66 (±0.13) | 90.07 (±1.68) | 78.05 (±0.33) | 73.54 (±1.01) |
| One For All | 83.74 (±0.09) | 75.93 (±0.12) | 81.05 (±0.34) | 73.42 (±1.39) | 63.17 (±0.21) | 66.65 (±0.82) | 75.21 (±0.23) | 69.08 (±0.64) | 72.29 (±0.12) | 60.35 (±1.25) | 82.52 (±0.13) | 87.64 (±1.98) | 76.33 (±0.19) | 72.18 (±1.03) |
| *Agentic Predictor* | **85.33** (±0.05) | **81.42** (±0.26) | **85.62** (±0.47) | **80.08** (±0.46) | **66.20** (±0.17) | **67.88** (±0.21) | **79.56** (±0.25) | **74.08** (±0.47) | **75.13** (±0.01) | **63.06** (±0.45) | **87.96** (±0.02) | 91.47 (±0.44) | **79.97** (±0.16) | **76.33** (±0.38) |
| Δ vs. best baseline (% Improvement) | +0.09 (0.11%) | +1.22 (1.52%) | +0.91 (1.07%) | +2.33 (3.00%) | +1.36 (2.09%) | +1.23 (1.85%) | +3.12 (4.08%) | +4.44 (6.38%) | +0.88 (1.19%) | +0.42 (0.67%) | +0.68 (0.78%) | -0.35 -(0.38%) | +1.61 (2.05%) | +2.79 (3.79%) |

**Evaluation Metrics**. To ensure a fair and consistent comparison, we strictly adhere to the official evaluation protocols specified by the benchmark.

- **Accuracy** quantifies how well a model predicts agentic workflow performance. It is defined as $accuracy = \frac{1}{|\mathcal{D}^{\text{test}}|} \sum_{i}^{|\mathcal{D}^{\text{test}}|} \mathbf{1}(\hat{e}_i = e_i)$, where $|\mathcal{D}^{\text{test}}|$ is the size of the test split, and $\hat{e}_i$ and $e_i$ denote the predicted and ground-truth performance, respectively. $\mathbf{1}(\cdot)$ is the indicator function, which returns 1 if $\hat{e}_i = e_i$, and 0 otherwise.
- **Utility** evaluates the consistency between the workflow rankings predicted by the model and the ground-truth rankings, emphasizing the model's ability to determine the relative order of different workflows. First, we calculate the ground-truth and predicted success rates of a workflow $\mathcal{W}_i$ by averaging $e$ and $\hat{e}$ across all tasks in $\mathcal{D}^{\text{test}}$. Then, we rank the workflows and extract the top-$k$ workflows according to the respective scores, resulting in two ordered sets: $\mathcal{H} = \{\mathcal{W}_i\}_{i=1}^{k}$ and $\hat{\mathcal{H}} = \{\mathcal{W}'_i\}_{i=1}^{k}$. Formally, $utility = \frac{1}{k} \sum_{i=1}^{k} \mathbf{1}(\mathcal{W}'_i \in \mathcal{H})$.

**Baselines**. Since there is no direct baseline method specifically designed for performance prediction in agentic systems, we adopt comparison baselines from the benchmark paper. Some of these methods have previously been used as performance predictors for NAS (White et al., 2021). The selected baselines include one naive **MLP** and several strong graph-based models: **GCN** (Kipf & Welling, 2017), **GAT** (Veličković et al., 2018), **GCN-II** (Chen et al., 2020), **Graph Transformer** (Shi et al., 2021), **Dir-GNN** (Rossi et al., 2024) and **One For All** (Liu et al., 2024a).

**Implementation Details**. For all methods, we follow the same setup as suggested by Zhang et al. (2025c). Specifically, we use a 2-layer backbone with a hidden dimension of 512, set dropout to 0.5, and use a batch size of 512. Models are optimized with the Adam optimizer (Kingma & Ba, 2014) using a learning rate of $1 \times 10^{-4}$ and weight decay of $5 \times 10^{-4}$. Training is conducted for 200 epochs on a single NVIDIA A100-SXM4-80GB GPU, and the best checkpoint is selected by the highest accuracy on the validation subset. Our framework is encoder-agnostic by design. To ensure a controlled comparison, we reuse the all-MiniLM-L6-v2 (Wang et al., 2020) text encoder and adopt CodeRankEmbed (Suresh et al., 2025) for code, both via SentenceTransformers (Reimers & Gurevych, 2019) with default hyperparameters. Text inputs are truncated to 256 tokens and encoded into 384-dimensional vectors, while code inputs (function-level nodes and full-workflow files) are tokenized and truncated to model limits (up to 8,192 tokens) producing 768-dimensional vectors. All embeddings are finally mapped into a unified 512-dimensional space using a 2-layer MLP ($L = 2$).

## 4.2 MAIN RESULTS (Q1)

We report all experimental results for agentic workflow performance prediction, averaged over three runs with different random seeds on the same dataset. Table 3 presents the main performance scores and standard deviations for all datasets. Our proposed framework, Agentic Predictor, consistently outperforms baseline methods across the three task domains. For **accuracy**, Agentic Predictor achieves top results in each domain—85.62%, 79.56%, and 87.96%, respectively—yielding the highest overall average accuracy of 79.97%. This corresponds to improvements of 2.05% to 6.90% over the comparison baselines. **Utility** scores show a similar pattern. Agentic Predictor attains the highest utility in code generation (81.42%) and math problem solving (74.08%), as well as

Table 4: Results of ablation study on different input view variations.

| Variations | | | Code Generation | | Math Problem | | Common Reasoning | | Average | |
|---|---|---|---|---|---|---|---|---|---|---|
| Code | Graph | Text | Accuracy | Utility | Accuracy | Utility | Accuracy | Utility | Accuracy | Utility |
| ✓ | | | 82.04 | 75.66 | 75.70 | 68.52 | 83.19 | **91.51** | 80.31 | 78.56 |
| | ✓ | | 84.44 | 77.22 | 79.14 | 67.99 | 87.00 | 91.03 | 83.53 | 78.75 |
| | | ✓ | 79.87 | 70.34 | 76.60 | 68.45 | 68.06 | 71.04 | 74.84 | 69.94 |
| ✓ | ✓ | | 83.72 | 73.97 | 75.86 | 70.18 | 86.88 | 86.14 | 82.15 | 76.76 |
| ✓ | | ✓ | 82.27 | 77.28 | 76.03 | 66.66 | 54.17 | 53.21 | 70.82 | 65.72 |
| | ✓ | ✓ | 82.45 | 74.64 | 75.70 | 67.83 | 69.47 | 70.55 | 75.87 | 71.01 |
| ✓ | ✓ | ✓ | **85.62** | **80.08** | **79.56** | **74.08** | **87.96** | 91.47 | **84.38** | **81.88** |

Table 5: Results of ablation study on different input graph variations.

| Variations | | Code Generation | | Math Problem | | Commong Reasoning | | Average | |
|---|---|---|---|---|---|---|---|---|---|
| Single View | Multi View | Accuracy | Utility | Accuracy | Utility | Accuracy | Utility | Accuracy | Utility |
| ✓ | | 82.58 | **78.52** | 78.57 | 67.51 | 86.95 | 90.14 | 82.70 | 78.72 |
| | ✓ | **84.44** | 77.22 | **79.14** | **67.99** | **87.00** | **91.03** | **83.53** | **78.75** |

a near-best score in reasoning tasks (91.47%), second only to GCN (91.82%). On average, it achieves the highest utility score of 76.33%, representing improvements of 3.79% to 5.87% over the baselines. These results demonstrate that Agentic Predictor not only enhances predictive accuracy but also improves downstream utility across diverse agentic workflows, highlighting its robustness and generalizability. The consistent performance gains further underscore the advantages of leveraging multi-view encoding for heterogeneous agentic workflows.

## 4.3 ADDITIONAL ANALYSES

**Ablation Study (Q2)**. To substantiate our contributions on specific design of multi-view workflow encoding in Agentic Predictor, we conduct ablation study on two main components using the AFlow subset: multi-view encoder and multi-graph encoding techniques. According to the results in Table 4, we find that incorporating all three input views—code, graph, and text—results in the best overall performance across all tasks. Specifically, the full model configuration achieves the highest average accuracy (84.38%) and utility (81.88%), underscoring the complementary value of each modality. Notably, the removal of any single view leads to a consistent drop in performance, demonstrating the synergistic role of multimodal inputs in prediction capabilities of Agentic Predictor.

Furthermore, results in Table 5 reveal the significance of multi-graph encoding. When multiple graphs are used instead of a single graph, the model shows a clear performance improvement, particularly in code generation (accuracy improves from 82.58% to 84.44%) and reasoning tasks (utility rises from 90.14% to 91.03%). This supports our hypothesis that different graph perspectives enrich structural context and lead to more robust representations. Together, these findings validate the architectural choices in Agentic Predictor, demonstrating that both multi-view and multi-graph designs are integral to its superior performance.

**Effects of Pretraining Phase (Q3)**. Since acquiring a large amount of ground-truth labels from agentic workflows is expensive, we examine whether cross-domain unsupervised pretraining (denoted as Agentic Predictor+) benefits settings where labeled instances are limited. We vary the label ratio from 0.1 to 0.5, selecting labeled samples from the training split of all datasets in the benchmark. We pretrain the proposed multi-view encoder on the remaining 50% ($M$) of the training set with a batch size of 32 for 20 epochs. See more details in §B.5. On average, the results shown in Figure 3 indicate that Agentic Predictor+ consistently outperforms all baseline models across all label ratios, demonstrating the effectiveness of our unsupervised pretraining strategy. The gains are especially

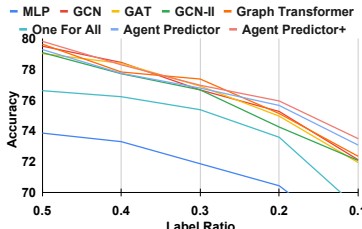

Figure 3: Accuracy comparison between Agentic Predictor and the baselines across varying label ratios.

pronounced in low-label regimes: at a 0.1 label ratio, Agentic Predictor+ maintains an accuracy above 73%, while other models drop closer to 70%. These findings underscore the importance of leveraging cross-domain structure through pretraining for generalizable workflow performance prediction, especially when direct supervision is limited.

**Out-of-Distribution (OOD) Performance (Q4)**. We evaluate OOD robustness under *cross-system* generalization (training on one agentic framework and testing on another) and *cross-domain* generalization (training on one task domain and testing on disjoint domains). As shown in §B.4, Agentic Predictor consistently generalizes beyond in-distribution memorization, maintaining strong performance and preserving relative workflow rankings across both settings. For example, when trained on AFlow and tested on G-Designer, Agentic Predictor improves average accuracy from 59.52% (best baseline) to 62.05% and utility from 55.33% to 58.49%. Similar gains hold in the reverse direction and under cross-domain splits.

**Comparison with LLM Predictors (Q5)**. We evaluate 5-shot, prompt-based LLM classifiers (temperature 0) using the standardized LLM-PP template (Jawahar et al., 2023) with GPT-4.1, Claude 4 Sonnet, and Gemini 2.5 Flash. As shown in Table 9, these prompt-only LLM predictors substantially underperform our graph-based model, indicating that they struggle to exploit the structured nature of agentic workflows. Agentic Predictor achieves 84.97% accuracy and 81.37% utility, far exceeding the second-best GPT-4.1 at 62.86% and 58.92%, while also avoiding the considerable latency and monetary overhead of LLM inference. Overall, few-shot LLMs serve as a useful baseline but remain less effective and less economical for large-scale agent search.

## 4.4 RESOURCE COST

We further examine the efficiency of Agentic Predictor measured by computation time and memory usage. As shown in Table 6, our framework remains competitive with standard GNN baselines despite its higher model capacity and richer input features, requiring only 0.054ms and 0.49GB to score a workflow at inference—orders of magnitude faster and cheaper than few-shot LLM predictors. A full run of Agentic Predictor involves a one-time cost of $\approx 1.2$ A100 GPU-hours (200 supervised epochs plus 20

Table 6: Computation cost comparison.

| Model | Training | | Inference | |
|---|---|---|---|---|
| | Time (s/epoch) | Memory (GB) | Time (ms/sample) | Memory (GB) |
| MLP | 0.195 | 0.033 | 0.002 | 0.020 |
| GCN | 4.867 | 0.058 | 0.017 | 0.040 |
| GAT | 5.108 | 0.058 | 0.023 | 0.042 |
| GCN-II | 4.623 | 0.058 | 0.015 | 0.040 |
| Graph Transformer | 5.372 | 0.087 | 0.023 | 0.060 |
| Dir-GNN | 4.965 | 0.077 | 0.023 | 0.050 |
| One For All | 6.140 | 0.038 | 0.018 | 0.038 |
| GPT-4.1 Claude 4 Sonnet Gemini 2.5 Flash | N/A (via OpenRouter API) | | 2253.333 1888.333 2606.667 | N/A |
| *Agentic Predictor* (+ pretraining) | 4.840 (168.104) | 2.760 (13.520) | 0.054 | 0.490 |

optional pretraining epochs), with modest memory requirements that fit on a single 16 GB GPU. In contrast, LLM-based scoring costs about $21 per 1,000 candidates ($\approx$$0.021 per sample with Gemini 2.5 Flash), implying a break-even point after only 110-120 evaluations assuming a $2/hr A100 rate. As realistic searches involve thousands of candidates and the trained predictor is reusable across tasks and frameworks, this modest one-time cost is quickly amortized, making Agentic Predictor far more economical than repeated LLM calls while offering near-zero marginal latency and higher accuracy.

Full experimental results on different underlying LLMs, various GNN backbones, LLM classifier comparison, and out-of-distribution test are reported in Tables 7 8, 9, 10, 11 and 12, respectively. An additional evaluation of performance predictors used as a reward function for agentic workflow optimization, and case study findings are also provided in §B.6 and §C.

## 5 CONCLUSIONS

This paper introduces Agentic Predictor, a novel framework for efficient prediction of agentic workflow performance that leverages a multi-view predictive approach. By integrating multi-view graph structures, code semantics, and prompt embeddings into a unified representation, Agentic Predictor captures the diverse characteristics of agentic systems. Moreover, it employs cross-domain unsupervised pretraining to mitigate the challenge of limited labeled data, thereby enhancing generalization across varied tasks. Through comprehensive experiments spanning three domains, Agentic Predictor consistently outperforms strong baselines in predictive accuracy and workflow utility.

**Limitations and Future Work**. While Agentic Predictor exhibits strong performance, it has certain limitations. The current predictor focuses on binary success metrics, constrained by the available benchmark, which may overlook more nuanced aspects of workflow behavior. Evaluating on new, independently curated agentic benchmarks is an important direction for future work. Additionally, adapting to highly specialized domains may still require some labeled data. Future work includes expanding to multi-objective optimization (e.g., balancing accuracy and cost), incorporating richer views such as temporal traces and user feedback, and exploring human-in-the-loop workflows for real-time refinement. These directions aim to make Agentic Predictor more generalizable and interactive in complex, real-world settings.

ACKNOWLEDGMENTS

This work was conducted as part of the research activities at DeepAuto.ai. This work was supported by Institute for Information & communications Technology Planning & Evaluation (IITP) grant funded by the Korea government (MSIT) (RS-2019-II190075, Artificial Intelligence Graduate School Program (KAIST)), IITP grant funded by MSIT (No.RS-2022-II220713, Meta-learning Applicable to Real-world Problems), Center for Applied Research in Artificial Intelligence (CARAI) grant funded by DAPA and ADD (UD190031RD), and National Research Foundation of Korea (NRF) grant funded by the Korea government (MSIT) (No. RS-2023-00256259).

## REPRODUCIBILITY STATEMENT

We facilitate reproducibility by providing an anonymous repository with all source code at `https://github.com/DeepAuto-AI/agentic-predictor`. Algorithm 1 provides the complete pseudocode of the proposed framework. For experimental consistency, the random seed for each run is $2^r$, where $r$ is the running index starting from 0.

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

## A  PSEUDOCODE OF AGENTIC PREDICTOR

We present the pseudocode of the proposed Agentic Predictor framework in Algorithm 1 below.

---
**Algorithm 1** Overall Procedure of Agentic Predictor

---
**Initialization:** Multi-View Encoder $\text{Enc}(\cdot)$ and Performance Predictor Model $\mathcal{M}_\Theta$
**Input:** User Instruction (or Task Description) $T \in \mathcal{T}$ and Training Data $D^{\text{train}}$
 1: ▷ Multi-View Graph Construction (§3.3)
 2: Construct a node-aligned view set $\mathcal{G} = \{\mathcal{G}_v = (\mathcal{V}, \mathcal{E}, X_v) \mid v \in \{\text{prompt}, \text{code}, \text{operator}\}\}$
    where $X_v$ is the view-specific node features.
 3: ▷ Cross-Domain Unsupervised Pretraining (§3.4, *Optional*)
 4: Sample $M$ unlabeled workflows $\mathcal{W}_1, \mathcal{W}_2, ..., \mathcal{W}_M$ from multiple domains
 5: **for** each $\mathcal{W}_i = (\mathcal{G}_i, \mathcal{C}_i, \mathcal{P}_i)$ **do**
 6:     $\mathbf{Z}_i \leftarrow \text{Enc}(\mathcal{G}_i, \mathcal{C}_i, \mathcal{P}_i)$                ▷ Encode multiview graph, code, and prompts
 7:     $(\hat{\mathcal{G}}_i, \hat{\mathcal{C}}_i, \hat{\mathcal{P}}_i) \leftarrow \text{Dec}(\mathbf{Z}_i)$                ▷ Decode reconstructions
 8: **end for**
 9: $\mathcal{L}_{enc} = \mathcal{L}_{rec} + \mathcal{L}_{con}$                ▷ Minimize total pretraining loss
10: ▷ Training Performance Predictor (§3.5)
11: Obtain (small) labeled dataset $\{(\mathcal{W}_j, T_j, e_j)\}_{j=1}^N$ from $D^{\text{train}}$
12: **for** each $(\mathcal{W}_j, T_j)$ **do**
13:     $\mathbf{Z}_j \leftarrow \text{Enc}(\mathcal{W}_j)$                ▷ Encode workflow
14:     $\mathbf{T}_j \leftarrow \text{TaskEncoder}(T_j)$                ▷ Encode task description
15:     $\mathcal{F}_j \leftarrow \text{MLP}([\mathbf{Z}_j, \mathbf{T}_j])$                ▷ Form joint representation
16:     $\hat{e}_j \leftarrow \mathcal{M}_\Theta(\mathcal{F}_j)$                ▷ Predict performance
17: **end for**
18: Train $\mathcal{M}_\Theta$ using binary cross-entropy loss $\mathcal{L}_{pred}(e_j, \hat{e}_j)$, where $\{e_j\}_{j=1}^N$
19: ▷ Predictor-Guided Candidate Ranking
20: Sample $K$ candidate workflows $\{\mathcal{W}_k\}_{k=1}^K$
21: **for** each $\mathcal{W}_k$ **do**
22:     $\mathbf{Z}_k \leftarrow \text{Enc}(\mathcal{W}_k)$                ▷ Encode workflow
23:     $\mathcal{F}_k \leftarrow \text{MLP}([\mathbf{Z}_k, \mathbf{T}])$                ▷ Encode task
24:     $\hat{e}_k \leftarrow \mathcal{M}_\Theta(\mathcal{F}_k)$                ▷ Predict score
25: **end for**
26: Rank all $\{\mathcal{W}_k\}$ by predicted scores $\hat{e}_k$
27: **return** top-$k$ ranked workflows for final evaluation

---

## B  ADDITIONAL EXPERIMENTAL RESULTS

This section provides complementary studies that further characterize our approach: robustness when the agent-controller LLM backbone varies (§B.1); an ablation over multiple GNN backbones (§B.2); a comparison to few-shot LLM predictors (§B.3); and out-of-distribution (OOD) generalization evaluations (§B.4).

### B.1  PERFORMANCE ON DIFFERENT LLM BACKBONES

As shown in Table 7, we assess whether predictor performance is robust when the agentic work-flows are driven by different LLMs. Concretely, we replicate our evaluation while swapping the controller LLM among GPT-4o-mini, DeepSeek, Qwen 7B, and Mistral 7B, holding the training data construction, multi-view encoder, and evaluation protocol fixed. Except for the Mistral 7B case, Agentic Predictor exhibits stable performance and preserves the relative ranking of workflows across these backbones, indicating that it captures structural and behavioral regularities of agentic programs rather than idiosyncrasies of any single LLM.

### B.2  PERFORMANCE ON DIFFERENT GNN BACKBONES

Our main experiments use a 2-layer GCN (hidden size 512) following the standard setup in FLORA-Bench (Zhang et al., 2025c), enabling a controlled comparison to baseline predictors. To test

Table 7: Results on different backbones driven agentic workflows.

| Domain | GPT-4o-mini | | DeepSeek | | Qwen 7B | | Mistral 7B | |
|---|---|---|---|---|---|---|---|---|
| Model | Accuracy | Utility | Accuracy | Utility | Accuracy | Utility | Accuracy | Utility |
| MLP | 83.88 | 76.16 | 85.89 | 71.72 | 84.25 | 80.52 | 89.23 | 85.07 |
| GCN | 82.94 | 80.40 | 86.56 | 75.69 | 86.71 | 84.48 | 92.58 | 88.48 |
| GAT | 83.03 | 80.25 | 84.42 | 75.18 | 86.98 | 84.26 | 92.62 | 88.72 |
| GCN-II | 82.81 | 79.48 | 84.34 | 75.68 | 85.17 | 82.71 | 90.94 | 85.89 |
| Graph Transformer | 83.42 | 79.83 | 86.34 | 73.06 | 86.76 | 84.65 | **92.80** | **88.87** |
| Dir-GNN | 84.85 | 79.81 | 85.38 | 71.27 | 86.36 | 84.50 | 91.87 | 88.47 |
| One For All | 81.24 | 71.92 | 84.73 | 73.23 | 84.51 | 80.42 | 89.13 | 85.06 |
| *Agentic Predictor* | **85.33** | **81.42** | **88.39** | **76.64** | **86.99** | **85.02** | 92.33 | 88.69 |

architecture sensitivity, we conduct an ablation over five diverse GNN backbones—GCN, GAT, GCN-II, Graph Transformer, and Dir-GNN—while keeping the prompt and code views fixed. As presented in Table 8 All backbones yield comparable predictive accuracy and replicate the same trends, reinforcing that the performance improvements stem from the multi-view encoding and pretraining rather than a specific GNN design. These results support the architecture-agnostic nature of the Agentic Predictor.

## B.3 COMPARISON WITH LLM PREDICTORS

We compare against few-shot, prompt-based LLM classifiers implemented with a standardized LLM-PP–style template (Jawahar et al., 2023) with 5-shot and temperature set to 0 using GPT-4.1, Claude 4 Sonnet, and Gemini 2.5 Flash. The results in Table 9 are consistent with prior findings on FLORA-Bench (Zhang et al., 2025c) (which evaluated DeepSeek-v3), these prompted LLMs underperform even a simple MLP predictor and substantially trail our graph-based approach. A likely reason is that prompted LLM classifiers do not exploit the structured execution patterns and tool-usage dynamics present in agentic workflows. Beyond accuracy, prompted LLM inference incurs a per-sample monetary and latency cost, whereas our predictor amortizes cost at training time. In our setup, generating predictions for up to 1,000 samples per task with LLM prompting required approximately $300, implying considerably higher expense at full-benchmark scale. By contrast, the learned predictor scales to large candidate sets with constant per-sample computational cost at inference. Overall, while few-shot LLMs provide a useful baseline, they are less effective and less economical for large-scale agent search.

## B.4 OUT-OF-DISTRIBUTION (OOD) GENERALIZATION PERFORMANCE

We study two factors that enable OOD robustness. First, the multi-view encoder jointly represents workflows via graph, code, and prompt views, all of which are architecture-agnostic. This design allows unseen agents and tools to be incorporated as long as their implementations and textual descriptions are available; the graph encoder embeds novel entities through structural and attribute signals without relying on fixed IDs. Second, cross-domain unsupervised pretraining over diverse unlabeled workflows equips the encoder with priors over common structural and behavioral motifs (e.g., tool invocation patterns and reasoning flows), improving robustness to unseen configurations.

Regarding evaluation, following RQ3 in FLORA-Bench (Zhang et al., 2025c), we perform two levels of OOD generalization. *Cross-system generalization:* train on one agentic framework (e.g., AFlow (Zhang et al., 2025b)) and test on another (e.g., G-Designer (Zhang et al., 2025a)) as well as *cross-domain generalization:* train on one set of downstream tasks (e.g., math) and test on disjoint tasks (e.g., coding) not observed during training. As presented in Table 10, Table 11 and Table 12, across both settings, Agentic Predictor maintains strong performance and preserves relative workflow rankings, indicating that it generalizes beyond in-distribution memorization.

## B.5 EFFECTS OF PRETRAINING PHASE (FULL RESULTS)

Since acquiring a large amount of ground-truth labels from agentic workflows is expensive, we examine whether cross-domain unsupervised pretraining (denoted as Agentic Predictor+) benefits settings where labeled instances are limited. We vary the label ratio from 0.1 to 0.5, selecting labeled samples from the training split of all datasets in the benchmark. Concretely, we construct an

Table 8: Results on different GNN backbones of Agentic Predictor.

| Domain | Code Generation | | Math Problem | | Common Reasoning | | Average | |
|---|---|---|---|---|---|---|---|---|
| GNN Backbone | Accuracy | Utility | Accuracy | Utility | Accuracy | Utility | Accuracy | Utility |
| GCN | 85.62 | 80.08 | 79.56 | 74.08 | 87.96 | 91.47 | 84.38 | 81.88 |
| GAT | 83.74 | 73.11 | 75.86 | 67.03 | 86.95 | 87.20 | 82.19 | 75.78 |
| GCN-II | 84.71 | 73.83 | 76.68 | 68.41 | 86.76 | 86.04 | 82.72 | 76.09 |
| Graph Transformer | 83.22 | 78.17 | 76.64 | 70.03 | 86.88 | 89.50 | 82.25 | 79.23 |
| Dir-GNN | 84.62 | 79.64 | 80.26 | 75.03 | 87.93 | 94.77 | 84.27 | 83.15 |

Table 9: Comparison between Agentic Predictor and LLM-based few-show classification.

| Domain | Code Generation | | Math Problem | | Common Reasoning | | Average | |
|---|---|---|---|---|---|---|---|---|
| Model | Accuracy | Utility | Accuracy | Utility | Accuracy | Utility | Accuracy | Utility |
| GPT-4.1 (∼$59) | 62.42 | 57.00 | 67.08 | 52.97 | 59.10 | 66.79 | 62.86 | 58.92 |
| Claude 4 Sonnet (∼$202) | 56.72 | 51.65 | 64.62 | 57.32 | 44.50 | 41.25 | 55.28 | 50.07 |
| Gemini 2.5 Flash (∼$21) | 60.52 | 58.94 | 51.60 | 55.21 | 59.20 | 63.17 | 57.10 | 59.11 |
| *Agentic Predictor* | **84.40** | **78.84** | **80.10** | **77.61** | **90.40** | **87.67** | **84.97** | **81.37** |

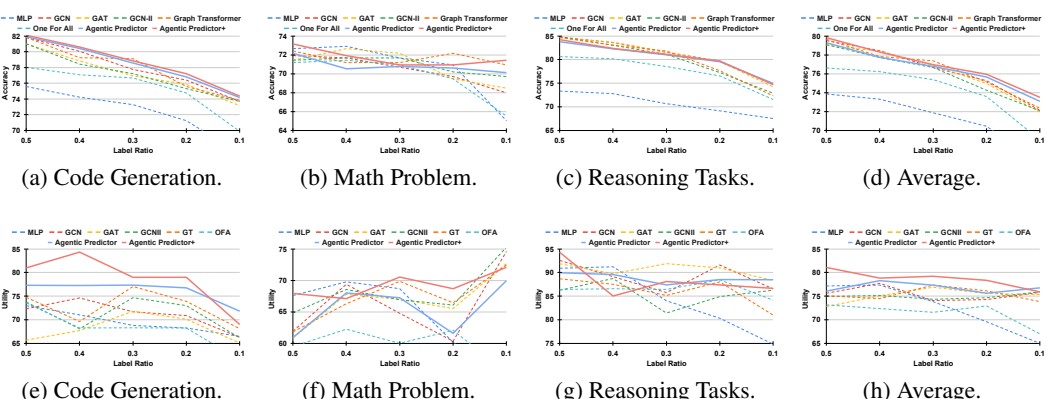

(a) Code Generation. (b) Math Problem. (c) Reasoning Tasks. (d) Average.

(e) Code Generation. (f) Math Problem. (g) Reasoning Tasks. (h) Average.

Figure 4: Comparison of accuracy (upper) and utility (lower) between Agentic Predictor and the baselines across varying label ratios.

unlabeled corpus by pooling all workflow configurations from the remaining training splits of the FLORA-Bench across Code, Math, and Reasoning tasks and both AFlow and G-Designer frameworks, sampling uniformly over the pool without additional deduplication or domain re-weighting. This yields $M = 232,104$ distinct samples (≈ 6.56% Code, 2.86% Math, 90.58% Reasoning). No validation and test workflows or labels are included to avoid leakage. We pretrain the proposed multi-view encoder with a batch size of 32 for 20 epochs.

Following the average results in the main text, we provide a comprehensive comparison of accuracy (top row) and utility (bottom row) across three task domains—code generation, math problems, and reasoning—under varying label ratios from 0.5 to 0.1 (Figure 4).

Across all settings, our proposed framework, Agentic Predictor, and its pretrained variant, Agentic Predictor+, consistently outperform baseline models, especially in low-resource scenarios. In the code generation domain (Figures 4a, 4e), Agentic Predictor+ achieves superior accuracy and notably higher utility as the label ratio decreases, outperforming all graph-based and non-graph baselines. Similarly, for math problems (Figures 4b, 4f), Agentic Predictor+ maintains a stable accuracy even as labeled data diminishes, while significantly improving utility, indicating better performance in label-scarce conditions. In reasoning tasks (Figures 4c, 4g), although accuracy deltas narrow between models, Agentic Predictor+ sustains strong utility across all label ratios, highlighting its robustness in generalization. When averaged across domains (Figures 4d, 4h), Agentic Predictor+ shows clear advantages in both metrics under limited supervision. The utility improvements are particularly prominent, suggesting that our pretrained encoder captures transferable representations that enhance decision-making, even when fine-tuning data is sparse. These findings validate the efficacy of the unsupervised pretraining phase and highlight the importance of cross-domain datasets for pretraining.

Table 10: Results when train on AFlow and test on G-Designer.

| Domain | Code Generation | | Math Problem | | Common Reasoning | | Average | |
|---|---|---|---|---|---|---|---|---|
| Model | Accuracy | Utility | Accuracy | Utility | Accuracy | Utility | Accuracy | Utility |
| GCN | 56.76 | 54.29 | 49.64 | 51.92 | 61.37 | 54.13 | 55.92 | 53.45 |
| GAT | 57.25 | 56.05 | 48.29 | 48.71 | 57.03 | 53.12 | 54.19 | 52.63 |
| GCN-II | 64.16 | 62.67 | 48.85 | 50.56 | 65.55 | 52.76 | 59.52 | 55.33 |
| Graph Transformer | 60.83 | 58.39 | 47.73 | 46.65 | 55.88 | 48.87 | 54.81 | 51.30 |
| One For All | 58.97 | 53.25 | 50.60 | 51.02 | 63.84 | 55.22 | 57.80 | 53.16 |
| *Agentic Predictor* | **65.02** | **64.91** | **53.62** | **52.83** | **67.51** | **57.74** | **62.05** | **58.49** |

Table 11: Results when train on G-Designer and test on AFlow.

| Domain | Code Generation | | Math Problem | | Common Reasoning | | Average | |
|---|---|---|---|---|---|---|---|---|
| Model | Accuracy | Utility | Accuracy | Utility | Accuracy | Utility | Accuracy | Utility |
| GCN | 58.21 | 57.33 | 67.57 | 54.63 | 57.51 | 53.37 | 61.10 | 55.11 |
| GAT | 59.29 | 59.68 | 66.34 | 52.70 | 56.07 | 50.38 | 60.57 | 54.25 |
| GCN-II | 58.75 | 61.17 | 67.32 | 52.96 | 55.93 | 52.19 | 60.67 | 55.44 |
| Graph Transformer | 60.52 | **61.44** | 58.97 | 57.49 | 56.50 | 54.86 | 58.66 | 57.93 |
| One For All | **62.01** | 54.57 | 58.72 | 61.23 | **59.40** | 54.17 | 60.04 | 56.66 |
| *Agentic Predictor* | 60.94 | 59.75 | **69.11** | **63.02** | 58.56 | **56.73** | **62.87** | **59.83** |

## B.6 WORKFLOW OPTIMIZATION RESULTS

In the main experiments, we demonstrate the feasibility and robustness of predicting agentic work-flow performance. However, it remains an open question whether such predictions can effectively contribute to improving efficiency and to what extent they may introduce performance degradation in agentic workflows. To investigate this, we evaluate *whether using Agentic Predictor as a predictor enhances the optimization of agentic workflows compared to alternative baselines*. Specifically, we measure the performance improvement (or loss) incurred when using performance predictors.

To ensure a fair comparison, we adopt the same experimental setup as Zhang et al. (2025c), which provides a unified platform for optimizing agentic workflows and evaluating their performance. During the optimization process on each benchmark, a predictor is used to estimate the performance of candidate agentic workflows. These predicted performance values are treated as rewards to guide the optimization. Upon completion of the optimization, the quality of the resulting workflows is assessed based on their accuracy score on held-out test tasks.

We compare Agentic Predictor against four baselines: (1) the **ground truth** baseline, which directly evaluates agentic workflows to obtain ground-truth performance scores (as done in the original AFlow (Zhang et al., 2025b)); (2) two strong GNN-based predictors **GCN** and **GAT**; and (3) a **random** baseline, which assigns random performance scores as rewards. This experiment is conducted across five benchmarks: MATH, GSM8K, MBPP, HumanEval, and MMLU.

As in Table 13, Agentic Predictor consistently outperforms the random, GCN, and GAT baselines, achieving an average accuracy score of **74.43%**, significantly higher than random (62.56%), GCN (68.42%), and GAT (71.00%). Notably, as a predictor incurs zero search cost compared to the ground-truth's cost of $39.83, this result underscores the effectiveness and efficiency of Agentic Predictor as a reliable predictor for optimizing agentic workflows. Note that the search cost is 0 because the predictors do not incur any LLM inference cost. Note that the search cost when using the performance predictor is effectively zero because the predictor incurs no LLM inference calls (i.e., no downstream task executions of task queries) to decide whether the current workflow has failed. The lightweight LLM update applied after each pass/fail decision, whose cost is about $0.005 - 0.01$ per update round with GPT-4.1-mini on the existing workflow, is *excluded* from the reported costs for all methods.

In real-world deployments, Agentic Predictor can be combined with any workflow generator (e.g., AFlow). In such cases, the overall cost decomposes into (1) the candidate-generation LLM cost (*shared across all search strategies*) and (2) the evaluation cost. Our predictor reduces (2) by replacing most candidate evaluations with cheap, yet more accurate predictions, while incurring only a one-time training cost (see §4.4). This analysis applies equally to the other predictors as well.

Table 12: Results on cross-domain OOD test.

| Domain | Code-Math | | Code-Reason | | Math-Reason | | Math-Code | | Reason-Code | | Reason-Math | | Average | |
|---|---|---|---|---|---|---|---|---|---|---|---|---|---|---|
| Model | Accuracy | Utility | Accuracy | Utility | Accuracy | Utility | Accuracy | Utility | Accuracy | Utility | Accuracy | Utility | Accuracy | Utility |
| GCN | 48.89 | 54.07 | 52.61 | 53.29 | 49.38 | 46.69 | 50.07 | 48.75 | 32.56 | 50.53 | 33.42 | 50.57 | 44.49 | 50.65 |
| GAT | 45.95 | 49.42 | 53.71 | 57.90 | 46.83 | 38.90 | 51.02 | 47.40 | 33.79 | 52.62 | 33.42 | 51.10 | 44.12 | 49.56 |
| GCN-II | 56.02 | 44.49 | 53.44 | 45.93 | 50.38 | 47.36 | 39.48 | 51.55 | 38.13 | 51.35 | 36.61 | **57.93** | 45.68 | 49.77 |
| Graph Transformer | 47.67 | 56.18 | 53.71 | 57.95 | 47.90 | 43.63 | 54.00 | 56.20 | 60.92 | 52.37 | 41.77 | 52.91 | 51.00 | 53.21 |
| One For All | 36.61 | **61.11** | 50.33 | 39.82 | 44.92 | 45.88 | **65.40** | 56.24 | **63.36** | 50.60 | 38.08 | 45.27 | 49.78 | 49.82 |
| *Agentic Predictor* | **57.17** | 61.03 | **54.22** | **62.99** | **53.86** | **61.75** | 59.88 | **60.25** | 61.60 | **54.52** | **62.90** | 52.69 | **58.27** | **58.87** |

Table 13: Workflow optimization performance based on the selected workflow across methods.

| Methods | Math Problems | | Code Generation | | Reasoning | | | Average | |
|---|---|---|---|---|---|---|---|---|---|
| | MATH | GSM8K | MBPP | HumanEval | MMLU | DROP | HotpotQA | Score | Search Cost ($) |
| Ground Truth (AFlow) | 87.38 | 94.53 | 73.22 | 97.20 | 83.10 | 84.25 | 69.94 | 84.23 | 39.83 |
| Random | 78.40 | 75.23 | 67.84 | 76.34 | 42.87 | 80.42 | 16.86 | 62.56 | 0.00 |
| GCN | 79.22 | 86.16 | 68.23 | 97.46 | 46.43 | 82.33 | 19.14 | 68.42 | 0.00 |
| GAT | 80.11 | 86.22 | **68.62** | 97.71 | 57.00 | 85.83 | **21.47** | 71.00 | 0.00 |
| *Agentic Predictor* | **81.89** | **92.65** | 68.42 | **98.73** | **79.70** | **86.25** | 13.37 | **74.43** | 0.00 |

## B.7 TRANSFERABILITY

Considering that the MMLU benchmark encompasses various reasoning tasks, we further investigate the transferability of predictors trained on MMLU datasets to determine whether they can be used to optimize similar reasoning tasks, specifically DROP (Dua et al., 2019) and HotpotQA (Yang et al., 2018). As reported in Table 13, the workflow optimized using Agentic Predictor achieves competitive performance on these tasks: 86.25% on DROP and 13.37% on HotpotQA, demonstrating notable transferability. While performance on HotpotQA is lower than the baselines, the results remain broadly comparable, indicating that the workflows optimized via Agentic Predictor maintain substantial effectiveness when transferred to closely related reasoning tasks. This highlights the practical potential of Agentic Predictor for broader applicability in workflow optimization scenarios.

## C   CASE STUDY

This section presents qualitative results from the workflow optimization process using Agentic Predictor as the reward function across three domains.

### C.1   CODE GENERATION

The code generation workflow on the HumanEval dataset demonstrates that the initial solution generation step often required subsequent refinement through explicit review and revision cycles. By systematically reviewing the initially generated code, and conditionally revising based on feedback from automated tests, the workflow substantially improved the final solution's correctness. This iterative approach effectively balanced computational cost and performance, resulting in solutions that were consistently more robust and accurate compared to single-step generations.

**Workflow for Code Generation (HumanEval)**

```python
from typing import Literal
import workspace.HumanEval.workflows.template.operator as operator
import workspace.HumanEval.workflows.round_19.prompt as prompt_custom
from metagpt.provider.llm_provider_registry import create_llm_instance
from metagpt.utils.cost_manager import CostManager

DatasetType = Literal["HumanEval", "MBPP", "GSM8K", "MATH", "HotpotQA", "DROP", "MMLU"]

class Workflow:
    def __init__(
        self,
        name: str,
        llm_config,
        dataset: DatasetType,
```

```
) -> None:
    self.name = name
    self.dataset = dataset
    self.llm = create_llm_instance(llm_config)
    self.llm.cost_manager = CostManager()
    self.custom = operator.Custom(self.llm)
    self.custom_code_generate = operator.CustomCodeGenerate(self.llm)
    self.test = operator.Test(self.llm)

async def __call__(self, problem: str, entry_point: str):
    """
    Implementation of the workflow
    1. Generate initial solution using custom_code_generate.
    2. Review the solution using custom operator.
    3. Test the solution; if test fails, revise using custom operator and retest.
    """
    # Step 1: Generate initial solution
    initial_solution = await self.custom_code_generate(problem=problem, entry_point=
        entry_point, instruction="")

    # Step 2: Review the solution to improve quality
    reviewed = await self.custom(input=initial_solution['response'], instruction=
        prompt_custom.REVIEW_PROMPT)

    # Step 3: Test the reviewed solution
    test_result = await self.test(problem=problem, solution=reviewed['response'],
        entry_point=entry_point)

    # If test fails, revise solution based on test feedback and retest once
    if not test_result['result']:
        revised = await self.custom(input=reviewed['response'] + "\n" + test_result['
            solution'], instruction=prompt_custom.REVISE_PROMPT)
        test_result = await self.test(problem=problem, solution=revised['response'],
            entry_point=entry_point)
        final_solution = revised['response'] if test_result['result'] else reviewed['
            response']
    else:
        final_solution = reviewed['response']

    return final_solution, self.llm.cost_manager.total_cost
```

## C.2  MATH PROBLEM

In addressing mathematical problems using the MATH dataset, the workflow leverages an ensemble strategy by producing multiple candidate solutions, subsequently selecting the most consistent one via a self-consistency ensemble step. The selected solution was then further refined through an additional review process. This combined ensemble and review mechanism significantly enhanced solution quality, highlighting the value of ensemble techniques in solving complex mathematical reasoning tasks, while maintaining a controlled computational budget.

**Workflow for Math Problem (MATH)**

```
from typing import Literal
import workspace.MATH.workflows.template.operator as operator
import workspace.MATH.workflows.round_88.prompt as prompt_custom
from metagpt.provider.llm_provider_registry import create_llm_instance
from metagpt.utils.cost_manager import CostManager

DatasetType = Literal["HumanEval", "MBPP", "GSM8K", "MATH", "HotpotQA", "DROP", "MMLU"]

class Workflow:
    def __init__(
        self,
        name: str,
        llm_config,
        dataset: DatasetType,
    ) -> None:
        self.name = name
        self.dataset = dataset
        self.llm = create_llm_instance(llm_config)
        self.llm.cost_manager = CostManager()
        self.custom = operator.Custom(self.llm)
        self.sc_ensemble = operator.ScEnsemble(self.llm)
```

```
async def __call__(self, problem: str):
    """
    Implementation of the workflow with ensemble and review step
    """
    # Generate multiple candidate solutions using custom operator with different
        instructions
    candidates = []
    for i in range(3):
        response = await self.custom(input=problem, instruction=prompt_custom.SOLVE_PROMPT
            + f" Attempt {i+1}.")
        candidates.append(response['response'])

        # Use self-consistency ensemble to select the best solution
        ensemble_result = await self.sc_ensemble(solutions=candidates, problem=problem)
        best_solution = ensemble_result['response']

        # Review and refine the best solution
        review_response = await self.custom(input=problem + "\nSolution to review:\n" +
            best_solution, instruction=prompt_custom.REVIEW_PROMPT)
        final_solution = review_response['response']

        return final_solution, self.llm.cost_manager.total_cost
```

## C.3 REASONING TASK

For reasoning tasks on the MMLU dataset, the workflow combines multiple generation techniques, including custom-generated solutions with varying prompts and answers produced by specialized answer-generation operators, to diversify initial candidate answers. The self-consistency ensemble step effectively selected the most consistent candidate, which was subsequently subjected to rigorous review and format verification steps. This meticulous process, which included conditional regeneration and revision to ensure strict adherence to specified answer formats, proved highly effective in enhancing both accuracy and reliability of the final responses.

**Workflow for Reasoning Task (MMLU)**

```
from typing import Literal
import workspace.MMLU.workflows.template.operator as operator
import workspace.MMLU.workflows.round_19.prompt as prompt_custom
from metagpt.provider.llm_provider_registry import create_llm_instance
from metagpt.utils.cost_manager import CostManager

DatasetType = Literal["HumanEval", "MBPP", "GSM8K", "MATH", "HotpotQA", "DROP", "MMLU"]

class Workflow:
    def __init__(
        self,
        name: str,
        llm_config,
        dataset: DatasetType,
    ) -> None:
        self.name = name
        self.dataset = dataset
        self.llm = create_llm_instance(llm_config)
        self.llm.cost_manager = CostManager()
        self.custom = operator.Custom(self.llm)
        self.answer_generate = operator.AnswerGenerate(self.llm)
        self.sc_ensemble = operator.ScEnsemble(self.llm)

    async def __call__(self, problem: str):
        """
        Implementation of the workflow with multiple custom answers, multiple AnswerGenerate
            answers, ensemble, review, and revision
        """
        # Step 1: Generate multiple candidate answers using custom operator with a concise
            prompt
        custom_answers = []
        for _ in range(2):
            custom_response = await self.custom(input=problem, instruction=prompt_custom.
                CUSTOM_PROMPT)
            custom_answer = custom_response['response']
            custom_answers.append(custom_answer)
```

```
        # Add 1 answer with diversity prompt to increase answer variety
        custom_diverse_response = await self.custom(input=problem, instruction=prompt_custom.
            CUSTOM_DIVERSE_PROMPT)
        custom_answers.append(custom_diverse_response['response'])

        # Step 2: Generate multiple candidate answers using AnswerGenerate operator to
            increase diversity
        answergen_answers = []
        for _ in range(2):
            answergen_response = await self.answer_generate(input=problem)
            answergen_answer = answergen_response['answer']
            answergen_answers.append(answergen_answer)

        # Step 3: Ensemble all candidate answers to select the most consistent answer
        all_answers = custom_answers + answergen_answers
        ensemble_response = await self.sc_ensemble(solutions=all_answers)
        ensemble_answer = ensemble_response['response']

        # Step 4: Review the ensemble answer to ensure format and correctness
        review_input = problem + "\nAnswer: " + ensemble_answer
        review_response = await self.custom(input=review_input, instruction=prompt_custom.
            REVIEW_PROMPT)
        reviewed_answer = review_response['response']

        # Step 5: If reviewed answer is not in correct format, regenerate with a stricter
            prompt
        if not reviewed_answer.startswith("Answer: Option "):
            strict_regen_input = problem + "\nPlease provide the final answer strictly in the
                format 'Answer: Option X'."
            strict_regen_response = await self.custom(input=strict_regen_input, instruction=
                prompt_custom.STRICT_REGEN_PROMPT)
            reviewed_answer = strict_regen_response['response']

        # Step 6: Revision step to refine the reviewed answer for strict format adherence
        revision_input = problem + "\nAnswer: " + reviewed_answer
        revision_response = await self.custom(input=revision_input, instruction=prompt_custom.
            REVISION_PROMPT)
        final_answer = revision_response['response']

        return final_answer, self.llm.cost_manager.total_cost
```

# D  USE OF LARGE LANGUAGE MODELS

In preparing this submission, we employed ChatGPT-5 strictly as a tool for language refinement, including polishing text, improving clarity, and correcting grammatical and typographical errors. Its role was limited to grammar correction, sentence restructuring, and rephrasing for readability. All model-generated content was thoroughly reviewed and revised by the human authors to ensure accuracy, originality, and adherence to research-integrity standards. The LLMs did not contribute to the core research ideas, experimental design, or any substantive intellectual components of the work. Note that LLMs also served as baselines for LLM-based prediction (§B.3) and case-study (§C) experiments, as described above.

