# OpenReview forum: "Multi-View Encoders for Performance Prediction in LLM-Based Agentic Workflows"
_ICLR.cc/2026/Conference — ICLR 2026 Poster_

### Official Review · Reviewer_R5e8 · 2025-10-15

**Soundness:** 4
**Presentation:** 3
**Contribution:** 4
**Rating:** 8
**Confidence:** 3

**Summary:**

The paper presents Agentic Predictor, a framework for predicting the performance of LLM-based agentic workflows without costly runtime evaluations. It introduces multi-view encoders that jointly model workflow graphs, code structure, and prompt semantics, combined with cross-domain unsupervised pretraining to address label scarcity. Experiments on the FLORA-Bench benchmark show consistent improvements in both prediction accuracy and workflow utility over existing graph-based baselines.

**Strengths:**

1. The idea of learning to predict workflow performance for LLM agents is interesting and timely, as it could make agent design more efficient.
2. The proposed multi-view representation is intuitively reasonable and well-motivated from both structural and semantic perspectives.
3. The experiments are technically thorough within the chosen benchmark, with clear ablations and implementation details that support reproducibility.

**Weaknesses:**

The architectural novelty is moderate; most encoder components are adapted from prior NAS and graph-representation literature, with limited methodological innovation beyond combining them.

**Questions:**

Can the authors clarify how multi-view information is balanced during aggregation? Is there any learned weighting or attention over views beyond simple concatenation?

---

> ### Author Response · Authors · 2025-11-24
> **Response to Reviewer R5e8**
>
> Thank you for your positive evaluation and for recognizing the relevance and practicality of predictor-based agent design. We hope the clarifications below further strengthen our paper and increase your confidence in its final assessment.
>
> > 1. Architectural Novelty.
>
> **R1.** We acknowledge that our encoder builds on established ideas from NAS and graph representation learning (as discussed in Performance Predictors for NAS, `Section 2`). Our contribution is not a new generic GNN architecture, but an adaptation of proven components to the **specific structure and heterogeneity of agentic workflows**. The novelty lies in *how* these components are integrated for this new problem setting.
>
> - A multi-view, node-aligned multigraph representation of workflows (structure, code, prompts) tailored to agent systems.
> - Node-level cross-view fusion that combines $\mathrm{CrossGraphAttn}$ with learned view weighting via $\mathrm{ViewAttnPool}$.
> - Demonstrated cross-framework robustness (AF ↔ GD), showing that the learned representation generalizes across real, heterogeneous agent platforms.
>
> While the individual components originate in prior NAS/GNN work, *their integration for agentic workflow performance prediction is, to our knowledge, **new** and **has not been studied** in prior literature*.
>
>
>
>
>
> > 2. How multi-view information is balanced during aggregation? Is there any learned weighting or attention over views beyond simple concatenation?
>
> **R2.** Yes, the aggregation is fully learned and attention-based rather than a fixed concatenation.
>
> First, in the **view-wise graph encoding** stage, each view-specific graph produces node embeddings $\mathbf{H}\_\text{prompt}$, $\mathbf{H}\_\text{code}$, and $\mathbf{H}\_\text{operator}$. These embeddings are stacked into $\mathbf{X} \in \mathbb{R}^{N \times V \times d}$ and processed by a multi-head $\mathrm{CrossGraphAttn}$ module, which **attends across views for each node**. Next, **learned view weighting** is performed by $\mathrm{ViewAttnPool}$. An MLP predicts per-node softmax attention weights over views, and a **weighted sum over views** yields fused node embeddings. Finally, a graph readout $G\_{\mathrm{pool}}$ then produces $\mathbf{Z}\_{\mathcal{G}}$, which we concatenate with the workflow-level prompt and code representations, $\mathbf{Z}\_{\mathcal{P}}$ and $\mathbf{Z}\_{\mathcal{C}}$, and pass through a **learnable MLP** to obtain the final workflow embedding $\mathbf{Z} = \mathrm{MLP}([\mathbf{Z}\_{\mathcal{G}}, \mathbf{Z}\_{\mathcal{C}}, \mathbf{Z}\_{\mathcal{P}}])$.
>
> We further highlight this mechanism in `Section 3.3.1`.

---

> > ### Comment · Reviewer_R5e8 · 2025-11-27
> >
> > The authors have addressed my questions regarding the architectural novelty by highlighting the specific adaptation to agentic workflows. Furthermore, the detailed explanation of the information fusion mechanisms resolves my query regarding how multi-view information is balanced. I maintain my original score.

---

### Official Review · Reviewer_MLNd · 2025-10-30

**Soundness:** 4
**Presentation:** 4
**Contribution:** 3
**Rating:** 6
**Confidence:** 3

**Summary:**

This paper proposes Agentic Predictor, a performance prediction framework for LLM-based agentic workflows. The core approach employs multi-view workflow encoding: graph view (topology and communication dependencies), code view (implementation, tool usage, and computational logic), and prompt view (system/instruction prompt semantics) are separately encoded and fused through an aggregation layer into a unified representation, followed by training a lightweight prediction head (MLP) for performance classification/regression on limited labeled pairs. In label-scarce scenarios, cross-domain unsupervised pretraining (reconstruction + contrastive learning) is adopted to learn generalizable representations, improving low-sample prediction performance. Experiments are conducted on FLORA-Bench subsets covering three domains: code generation, math problem solving, and commonsense reasoning, reporting consistent improvements in accuracy and utility, along with robustness across different LLM/GNN backbones and OOD scenarios. Ablations show that both multi-view and multi-graph encoding contribute gains, with pretraining being particularly effective under low label ratios.

**Strengths:**

1. **Novel and well-motivated approach**: Decomposes agentic workflows into three complementary views (graph, code, prompt), balancing structural dependencies and semantic signals, fitting the "heterogeneous, label-scarce" problem nature.
2. **Effective pretraining strategy**: Cross-domain unsupervised pretraining significantly improves prediction quality in low-label scenarios, offering practical sample efficiency advantages.
3. **Efficiency and practicality**: The lightweight predictor serves as a candidate ranker, substantially reducing expensive LLM calls, achieving a "search-agnostic, composable" accelerator paradigm.
4. **Robustness and generality**: Maintains good ranking consistency and accuracy across different LLM/GNN backbones and cross-system/cross-task OOD settings.
5. **Comprehensive experiments**: Main results + view/multi-graph/pretraining ablations + workflow optimization + case studies provide thorough coverage and clear argumentation.

**Weaknesses:**

**1. Missing Multi-graph Definition**

- The paper uses G_prompt, G_code, G_operator fused via CrossGraphAttn/ViewAttnPool, but does not clarify how the three graphs are constructed: whether they are isomorphic (sharing V, E), the source and representation of each graph's node/edge features (text embeddings, AST/CFG, operator types, etc.), how they are specifically generated from W={V,E,P,C}, and the boundary with FLORA-Bench "following". The absence of formal definitions/pseudocode/illustrations directly impacts reproducibility and contribution assessment.

**2. Insufficient Unsupervised Pretraining Data Description**

- The paper does not specify the scale of unsupervised pretraining data M, source/domain composition, sampling and deduplication strategies, whether there is overlap with test domains/frameworks, and measures to prevent potential label/semantic leakage. These factors directly relate to the credibility and transferability of +pretraining.

**3. Code Encoder Too Black-box**

- Missing details on input representation (specific usage of CodeRankEmbed, whether combined with AST/CFG/call graphs), length normalization/truncation strategies. As the code view is critical to final performance, design details should be supplemented.

**4. Incomplete Cost Analysis**

- While emphasizing zero LLM cost during search, the paper lacks detailed breakdown of one-time training/pretraining time/memory/dollar costs, and comparisons with few-shot LLM classifiers at different scales regarding cost inflection points.

**Questions:**

1. **Multi-graph construction**: Please formally define G_prompt/G_code/G_operator (whether isomorphic, node/edge features, construction process from W with pseudocode/illustrations).
2. **Unsupervised pretraining data details**: Scale of M, domain distribution, sampling and deduplication, strategies to prevent overlap with test domains/frameworks; if cross-framework (AFlow/G-Designer) or cross-domain mixing exists, please clarify the partitioning and anti-leakage measures.
3. **CodeRankEmbed implementation details**: How does the code view convert source code to vectors (tokenization/truncation/aggregation, whether combined with AST/CFG/call graphs), and why not choose some large-scale pretrained general-purpose embedding models?
4. **Cost clarification**:
   - Explain the basis for "search phase LLM cost is 0" in the paper (is it because candidates come from existing workflow pools rather than LLM generation?).
   - If LLM generation of candidates is needed in real systems, how would the overall cost change? You may discuss this appropriately.

---

> ### Author Response · Authors · 2025-11-24
> **Response to Reviewer MLNd (1/2)**
>
> We thank the reviewer for the positive and detailed assessment. We hope the following clarifications strengthen our paper and further increase your confidence in the final evaluation.
>
> > 1. Missing Formal Multi-Graph Definition.
>
> **R1.** In the revised manuscript, we provide formal definitions of the three graph views and pseudocode for constructing them from a workflow $\mathcal{W} = \\{\mathcal{V}, \mathcal{E}, \mathcal{P}, \mathcal{C}\\}$ in `Section 3.3.1` (Graph Encoder) and `Algorithm 1`. Specifically,
> - $\mathcal{G}\_{\text{prompt}}$: node features are pooled embeddings of each agent's system and instruction prompts.
> - $\mathcal{G}\_{\text{code}}$: node features are the function-call-level code associated with each agent.
> - $\mathcal{G}\_{\text{operator}}$: node features encode operator types, definitions, and resource hints.
>
> All three graphs share the same node set and edge set, where edges are defined using the Abstract Syntax Tree (AST)–based construction directly provided by the benchmark. Each graph view is encoded with a GNN to produce $\mathbf{H}\_{\text{prompt}}$, $\mathbf{H}\_{\text{code}}$, and $\mathbf{H}\_{\text{operator}}$, which are stacked along a view dimension and fused via the $\mathrm{CrossGraphAttn}$ and $\mathrm{ViewAttnPool}$ modules before graph-level readout. These additions make the multi-graph construction explicit and address the reviewer's concerns about missing formal definitions and reproducibility.
>
>
> > 2. Insufficient Unsupervised Pretraining Data Description.
>
> **R2.** Our pretraining procedure **does not leak any labels from the validation or test sets**. The benchmark provides disjoint train, validation, and test splits, and we strictly adhere to this separation throughout all experiments. As updated in `Sections 4.3` and `B.5`, we vary the labeled ratio from 0.1 to 0.5 by sampling labeled examples only from the *training* split of each dataset in the benchmark.
>
> During pretraining, we use a cross-domain corpus of **unlabeled workflows drawn *exclusively* from the training splits** across all Code:Math:Reasoning × GD:AF datasets. No validation or test workflows, and no performance labels or templates, are used at this stage. Agentic Predictor+ therefore benefits from cross-domain structural information **without any leakage** of evaluation labels or test-time templates. To keep the setup simple and reproducible, we do not apply any additional deduplication beyond the dataset splits defined originally. Each (workflow configuration, dataset, framework) triple in the training splits is treated as a separate unlabeled example. We randomly shuffle this corpus at the beginning of pretraining and draw mini-batches without replacement within each epoch.
>
> Finally, this pretraining phase is specifically designed to address label scarcity and is conceptually distinct from the cross-system (or cross-domain) transferability experiments in `Section B.4`, which train on one system (or domain) and evaluate on another *without* any pretraining.
>
> > 3. Code Encoder and CodeRankEmbed Implementation Details
>
> **R3.** As updated in `Section 4.1` (Implementation Details), our framework is encoder‑agnostic by design. In our implementation, we instantiate both text and code encoders via the Sentence Transformers library and keep all encoder hyperparameters at their default values and tokenizers to isolate the contribution of our multi-view formulation and avoid confounding factors. Below we make these choices explicit.
>
> Prompt encoder uses `all-MiniLM-L6-v2` (256-token max length and 384-dim embeddings). Code encoder uses `nomic-ai/CodeRankEmbed` (8192-token max length and 768-dim embeddings). CodeRankEmbed operates only on raw source code either function-level per node for graph or file-level for the code encoder. We choose CodeRankEmbed because it is purpose-built for code and outperforms large general encoders on code retrieval benchmarks, while keeping our comparison focused on multi-view modeling rather than encoder scale.
>
> Note that the newly proposed graph views, $\mathcal{G}\_{\text{code}}$ and $\mathcal{G}\_{\text{operator}}$, capture structural information from AST, CFG, and call graphs. The separate Code Encoder is therefore designed to **capture a holistic view of the entire code snippet, rather than focusing on specific function calls**, which are already modeled by the graph views. We have revised `Sections 3.3.1` and `4.1` to clarify these points.

---

> ### Author Response · Authors · 2025-11-25
> **Response to Reviewer MLNd (2/2)**
>
> > 4. Incomplete Cost Analysis.
>
> **R4.** We now include a detailed cost analysis in `Section 4.4` and `Table R4`. The revised manuscript reports both the one-time (pre-)training cost of the predictor and its inference-time cost during search. Using the per-epoch measurements in `Table R4` (`Table 6` in the revised paper) and our full training schedule (200 supervised epochs + 20 pretraining epochs), the total wall-clock training time for Agentic Predictor+ is approximately 4,330 seconds (≈ 1.2 GPU-hours), with peak memory usage of 13.52 GB, which comfortably fits on a single 16 GB GPU.
>
>
> **Table R4:** Computation Cost Comparison. `n/a` indicates that the calculation is not applicable because the OpenRouter API was used.
> |               **Model**              |    **Training**    |                   |     **Inference**    |                 |
> |:------------------------------------:|:------------------:|:-----------------:|:--------------------:|:---------------:|
> |                                      | **Time (s/epoch)** |  **Memory (GB)**  | **Time (ms/sample)** | **Memory (GB)** |
> |                  MLP                 |        0.195       |       0.033       |         0.002        |      0.020      |
> |                  GCN                 |        4.867       |       0.058       |         0.017        |      0.040      |
> |                  GAT                 |        5.108       |       0.058       |         0.023        |      0.042      |
> |                GCN-II                |        4.623       |       0.058       |         0.015        |      0.040      |
> |           Graph Transformer          |        5.372       |       0.087       |         0.023        |      0.060      |
> |                Dir-GNN               |        4.965       |       0.077       |         0.023        |      0.050      |
> |              One For All             |        6.140       |       0.038       |         0.018        |      0.038      |
> |                GPT-4.1               |         n/a        |        n/a        |         2253.333        |       n/a       |
> |            Claude 4 Sonnet           |         n/a        |        n/a        |         1888.333        |       n/a       |
> |           Gemini 2.5 Flash           |         n/a        |        n/a        |         2606.667        |       n/a       |
> | *Agentic Predictor* (+ pretraining) | 4.840 (168.104) | 2.760 (13.520) |         0.054        |      0.490      |
>
>
> For comparison, the cheapest 5-shot LLM-based prediction in our LLM-PP setup (i.e., Gemini 2.5 Flash) costs roughly \\$21 per 1,000 predictions, i.e., about \\$0.021 per sample. If we denote the hourly GPU price by $p$, the one-time training cost of Agentic Predictor+ is $1.2p$ dollars, while the marginal cost of an LLM-only classifier is \\$0.021 per evaluation. Equating these, $1.2p=0.021N⇒N≈57p$, so for a realistic A100-class price of $p$ ≈ \\$2/GPU-hour on modern GPU clouds [a], the **break-even point** (inflection points) is only $N≈57×2≈114$ workflow evaluations. In other words, after on the order of a hundred candidate evaluations, the one-time training cost of the predictor is already amortized, and for the hundreds of thousands of workflows considered in our experiments (see # samples in `Table 2`) and typical real-world search regimes, the predictor's up-front cost is negligible relative to repeated LLM calls. Furthermore, `Table R4` shows that the predictor’s inference speed is 0.054ms per sample, orders of magnitude faster than LLM inference, and that its inference memory footprint (≈0.49GB) is tiny compared to running an LLM. Thus, in realistic usage, Agentic Predictor delivers consistently lower marginal latency and monetary cost than few-shot LLM classification, while also achieving higher accuracy.
>
> Finally, the monetary **search cost** when using the performance predictor is effectively zero, because the predictor incurs no LLM inference calls (i.e., no downstream task executions of task queries) to decide whether the current workflow has failed. For fairness, the lightweight LLM update applied after each pass/fail decision, whose cost is about \\$0.005-\\$0.01 per update round with GPT-4.1-mini on the existing workflow, is *excluded* from the reported costs for all methods. In real-world deployments, Agentic Predictor can be combined with any workflow generator (e.g., AFlow). In such cases, the overall cost decomposes into (1) the candidate-generation LLM cost (***shared across all search strategies***) and (2) the evaluation cost. Our predictor reduces (2) by replacing most candidate evaluations with cheap, yet more accurate predictions, while incurring only a one-time training cost. This analysis applies equally to the other predictors as well.
>
> [a] https://gpus.io/gpus/a100 (average price around \$2.03)

---

> > ### Comment · Reviewer_MLNd · 2025-11-25
> >
> > I thank the authors for their detailed response. In particular, the added formal definitions and the comprehensive cost analysis have effectively addressed my main concerns.
> >
> > I have also carefully read the comments from the other reviewers. Given these clarifications and my original assessment, I will maintain my current rating.

---

> > > ### Author Response · Authors · 2025-11-26
> > >
> > > Dear Reviewer MLNd,
> > >
> > > We sincerely appreciate your detailed feedback, which has helped us substantially improve the clarity and completeness of the paper. We are glad that the revisions clarified the issues raised.
> > >
> > > Thank you for your careful evaluation and for confirming that the additional clarifications effectively addressed your main concerns.
> > >
> > >
> > > Best regards,
> > >
> > > The Authors

---

### Official Review · Reviewer_h3gr · 2025-10-31

**Soundness:** 2
**Presentation:** 2
**Contribution:** 2
**Rating:** 4
**Confidence:** 2

**Summary:**

The authors present a system to predict how well different workflows of agents built on large language models (LLMs) will perform, before actually running them. They encode workflows using three “views” (structure/graph of agents, code logic, and prompts) and pre-train unsupervised across domains, then fine-tune for prediction of success/utility. The method is inspired by recent work in Neural Architecture Search task that effectively evaluates architectures. They show that this multi-view approach improves prediction accuracy and helps guide workflow design with fewer expensive evaluations.

**Strengths:**

- The motivation (too many workflows, high cost to evaluate) is convincing.
- The multi-view encoding idea is intuitive and makes sense: workflows are complex, so capturing various facets (graph, code, prompt) is a logical approach.
- The pre-training across domains is a good practical touch: many real-world tasks have limited labels, so this should help generalization.

**Weaknesses:**

In Table 3, it is not clear how the improvement percentages are computed. When comparing Agentic Predictor to the best baseline, the gains appear much smaller. It seems that the authors calculate improvements relative to the simplest baseline (MLP), which could be misleading. Are the other rows in table 3 from the author? Is the MLP the previous state of the art baseline?
Overall, the paper tends to overstate the magnitude of the improvements. The results are positive but more modest than implied.

Minor comments:
In Table 2, please define the abbreviations GD and AF directly in the caption for clarity.

**Questions:**

see weaknesses

---

> ### Author Response · Authors · 2025-11-25
> **Response to Reviewer h3gr**
>
> We appreciate the reviewers' constructive comments regarding the clarity of the reported results and their interpretation. We hope the following clarifications strengthen our submission and increase your confidence in its final evaluation.
>
> > 1. Improvement percentages and SOTA status.
>
> **R1.** We acknowledge that the original “% Improvement (Up to)” row and the unclear baseline specification could lead to misinterpretation. We have therefore revised `Table 3` and the accompanying discussion. All gains in `Table 3` are now computed relative to the **second-best method** in each column, and the table explicitly reports “Δ vs. best baseline” with both absolute and relative improvements.
>
> Due to page limitations, we previously omitted standard deviations. With the additional space now available, we report mean ± standard deviation over three random seeds. These results show that the improvements, although modest, are **consistent and statistically meaningful**.
>
>
> We also clarify that MLP serves only as a simple non-graph reference baseline rather than a prior SOTA method. All graph baselines (e.g., GCN and GAT) are our own re-implementations following the standardized FLORA-Bench protocol (`Section 4.1`). We believe these revisions improve transparency, strengthen the validity of the comparisons, and reduce the risk of misleading interpretations.
>
>
>
> > 2. Define GD and AF in table captions.
>
> **R2.** As suggested, we now define GD (G-Designer) and AF (AFlow) in `Section 4.1` and in the table caption. We use these abbreviations consistently throughout the paper.

---

### Official Review · Reviewer_rPuy · 2025-11-01

**Soundness:** 3
**Presentation:** 3
**Contribution:** 2
**Rating:** 4
**Confidence:** 4

**Summary:**

This paper proposes Agentic Predictor, a performance prediction framework for LLM-based agentic workflows that aims to reduce expensive trial-and-error evaluations during workflow optimization. The core technical contribution is a multi-view encoding scheme that integrates graph structures, code semantics, and prompt embeddings, combined with optional cross-domain unsupervised pretraining to address label scarcity. Experiments on FLORA-Bench show improvements of up to 12.12% in prediction accuracy and 15.16% in workflow utility over baseline methods across three domains (code generation, math, and reasoning tasks).

**Strengths:**

The paper addresses a practically important challenge—reducing the computational cost of evaluating agentic workflows—and clearly articulates why existing execution-based approaches are inefficient for workflow optimization. The evaluation is thorough, including ablation studies, low-label regime analysis, OOD generalization tests, and workflow optimization experiments, demonstrating the predictor's effectiveness across multiple dimensions. The method consistently outperforms strong GNN-based baselines (GCN, GAT, Graph Transformer, etc.) with notable margins, and the pretrained variant (Agentic Predictor+) shows clear advantages in label-scarce scenarios.

**Weaknesses:**

1. The core contribution essentially combines existing techniques (multi-graph GNN, cross-view attention, contrastive pretraining) without fundamental innovation, the multi-view encoding is a relatively straightforward ensemble of three modality-specific encoders, and the pretraining strategy follows standard contrastive + reconstruction objectives commonly used in multi-modal learning.
2. All experiments rely on a single benchmark (FLORA-Bench) with its specific workflow representation format; it remains unclear whether the design choices (three specific views, multi-graph encoding, cross-view attention) would transfer to other agentic frameworks with different architectures, and the OOD tests are still within the same benchmark ecosystem rather than truly novel workflow paradigms.

**Questions:**

As shown in weakness.

---

> ### Author Response · Authors · 2025-11-25
> **Response to Reviewer rPuy**
>
> We appreciate the reviewer's positive assessment of the work's practical relevance and extensive experimentation. Below, we address the two concerns.
>
>
> > 1. Combination of existing techniques; modest methodological innovation.
>
> **R1.** We acknowledge that our model builds on established architectural primitives. However, our contribution lies in ***how** these components are recomposed into, to our knowledge, **the first workflow-specific, multi-view predictor tailored to LLM-based agentic systems***, which differ fundamentally from single-view graph architectures and from the GNN baselines in FLORA-Bench.
>
> Concretely, our paper introduces (1) multi-view graph decomposition that provides complementary relational information; (2) learnable node-wise $\textrm{CrossGraphAttn}$ and $\textrm{ViewAttnPool}$ attention modules that estimate per-node, per-view importance and focus on the most informative modalities; (3) non-graph view encoders on prompt and code views that capture workflow holistic properties beyond architectural aspects; and (4) cross-domain pretraining that improves efficiency in label-scarce regimes.
>
> **None of these design choices** are present in FLORA-Bench, which is the only prior work addressing a similar research question. Ablation studies (`Tables 4-5`) substantiate the significance of these mechanisms, and `Table 1` summarizes the high-level distinctions between our approach and recent studies.
>
> > 2. Evidence beyond a single benchmark/representation format.
>
> **R2.** We agree that broader benchmarks would be valuable. However, FLORA-Bench (*the only publicly available benchmark for this problem*) was explicitly designed with cross-framework diversity in mind. It already includes **two *independently* developed** agentic frameworks, AFlow and G-Designer, which differ in architecture, codebase, tool ecosystem, and workflow representation. Thus, even within this benchmark, evaluation naturally spans heterogeneous system designs, directly addressing the concern.
>
> Our experiments leverage this diversity. As shown in `Tables 9-10`, our predictor consistently outperforms all graph-based baselines on average, across these heterogeneous workflow systems, not just across workflows within a single framework.
>
> `Table 11` further evaluates cross-domain generalization by training on one domain (e.g., Code) and testing on others (e.g., Math/Reasoning). The model maintains strong performance under these distributional shifts, indicating robustness to both task-type and domain changes.

---

### Author Response · Authors · 2025-11-25
**Response and Revision Summary**

We thank all reviewers for their thoughtful and constructive feedback. We appreciate the recognition of our **clear motivation** (h3gr, MLNd, R5e8), the **strength of the multi-view design** (h3gr, R5e8), the **benefits of cross-domain pretraining** (h3gr, MLNd), and the **comprehensiveness of our experimental evaluation** across domains, backbones, and OOD settings (rPuy, MLNd, R5e8).

Our rebuttal provides clarifications on:
1. Novelty and conceptual positioning of our method
2. Formal definition of the multi-graph encoder (revised `Section 3.3.1`)
3. Unsupervised pretraining protocol (revised `Section 4.3` and `Section B.5`)
4. Implementation details for the code-view encoder (expanded `Section 4.1`)
5. Cross-framework generalization analysis (revised `Section 4.3`, text reorganized from `Section B.4`)
6. Accurate reporting and cost analysis (updated `Table 3`, `Section 4.2`, and new `Section 4.4`)

To the best of our knowledge, this paper is the first work to propose a performance prediction framework *specifically tailored* to LLM-based agentic workflows. While FLORA-Bench provides several GNN baselines, these use generic graph encoders and do ***not*** introduce a predictor designed around a multi-view representation of agentic workflows. **This contextualizes our contribution** and explains why the baselines in our study are our own re-implementations.

We also emphasize that the benchmark we use (currently the only publicly available one) includes **two *independently* implemented agentic frameworks** (AFlow and G-Designer), allowing us to demonstrate genuine cross-system transfer under different underlying configurations.

With these clarifications and additional details, we believe the rebuttal and the revised paper (with changes highlighted in **brown**) now more clearly communicate the contributions and practical impact of our work, and we hope they address the remaining concerns.

---

### Author Response · Authors · 2025-12-01
**Post-Rebuttal Summary for Area Chair**

Dear Area Chair and Reviewers,

We appreciate the opportunity to provide a concise summary of how our rebuttal and revisions address the key concerns, especially given that the discussion period ended earlier than scheduled. As a result, **two reviewers (rPuy, h3gr) were unable to provide any follow-up reply to our responses** even though their concerns are fully addressed in the revised manuscript.

The main issues fall into three themes: (1) conceptual and architectural novelty (R5e8, rPuy), (2) reliance on a single benchmark (rPuy), and (3) clarity of reported gains, implementation details, and cost analysis (h3gr, MLNd).

1. **Novelty**. Our revised explanation clarifies that *Agentic Predictor* is, to our knowledge, ***the first workflow-specific, multi-view predictor for LLM-based agentic systems***, with node-aligned multigraph views and learned cross-view fusion. **Reviewer R5e8 explicitly acknowledges this clarification**. Given that Reviewer rPuy raised the same concern, and our response directly parallels what Reviewer R5e8 found satisfactory, it is reasonable to infer that Reviewer rPuy would likely have updated their assessment had they been able to follow-up.

2. **Benchmark Diversity**. Regarding the single benchmark concern, we emphasize that FLORA-Bench is currently the *only publicly available benchmark* for this problem and that it was intentionally designed to include **two *independently implemented* agentic frameworks** (AFlow and G-Designer). Our cross-framework and cross-domain analyses demonstrate that the design generalizes across heterogeneous architectures, directly addressing the reviewer's concern.

3. **Clarity and Cost Analysis**. We revised `Table 3` to clearly report improvements relative to the strongest baseline with standard deviations, clarified implementation details, and added a full training- and inference-cost analysis. **Reviewer MLNd confirms that the main concerns have been resolved**. Reviewer h3gr's concern on result presentation is fully addressed through the improved tables and definitions, though they were unable to follow-up due to the early cutoff.


The table below provides the overview summary.
| Reviewer | Main Concern(s) | Resolution in Revision | Follow-up |
|:-------:|------------------|------------------------|-----------|
| rPuy    | Novelty; single benchmark | Clarified workflow-specific design; emphasized cross-framework/domain evidence | *Unable to follow-up* (early cutoff) |
| h3gr    | Result presentation clarity | Revised `Table 3`; added definitions and clearer baselines | *Unable to follow-up* (early cutoff) |
| MLNd    | Multi-graph definitions; pretraining details; code encoder; cost analysis | Added formal definitions, data descriptions, cost table | **Confirmed resolved**; *retained* score 6 |
| R5e8    | Moderate novelty; fusion mechanism | Clarified adaptation and learned cross-view weighting | **Confirmed resolved**; *retained* score 8 |


All revisions are fully integrated and highlighted in $\color{brown}{\textbf{brown}}$. We hope this summary helps clarify the final state of the manuscript and the expected reviewer perspectives had the discussion period proceeded normally. Thank you once again for your time and consideration. If the new Area Chair has any remaining concerns or questions raised by the reviewers, we would be happy to address them.


Best regards,

The Authors

---

### Meta-Review · Area_Chair_3Ykf · 2026-01-09

**Summary:**

This paper proposes Agentic Predictor, a lightweight framework for predicting the performance of LLM-based agentic workflows without expensive runtime evaluation. The core contribution is a multi-view workflow encoding that integrates structural (graph), code, and prompt views, combined with cross-domain unsupervised pretraining to address label scarcity. Reviewers agreed that the problem is timely and practically important, and that the approach is well motivated. Experiments on FLORA-Bench demonstrate consistent improvements in prediction accuracy and workflow utility across multiple domains, frameworks, and out-of-distribution settings. I recommend acceptance.

**Reviewer Concerns:**

Reviewers raised concerns primarily around moderate architectural novelty, reliance on a single benchmark, and clarity of reporting and implementation details. These concerns were substantially addressed in the rebuttal and revised manuscript. The authors clarified the workflow-specific novelty of the multi-view, node-aligned multigraph design; provided formal definitions and pseudocode for graph construction; expanded details on the unsupervised pretraining data and protocol; clarified the code-view encoder design; and added a comprehensive cost analysis. Result presentation was revised to report improvements relative to the strongest baselines with standard deviations, improving transparency. While the methodological components build on existing ideas, reviewers agreed that their integration for agentic workflow performance prediction is novel and effective, and the empirical validation is thorough within the constraints of the available benchmark.

**Reviewer Scores:**

Reviewer scores ranged from marginal to strong accept, with one clear accept recommendation and others indicating they would not object to acceptance. Importantly, reviewers who raised substantive concerns confirmed that these were resolved after clarification, and no reviewer maintained a strong objection.

---

### Decision · Program_Chairs · 2026-01-26

Accept (Poster)